biomechanics, evolution

baculum, carnivore, sexual selection, shape complexity, evolution, genital

**Author for correspondence:**
Charlotte A. Brassey
e-mail: c.brassey@mmu.ac.uk

# Postcopulatory sexual selection and the evolution of shape complexity in the carnivoran baculum

Charlotte A. Brassey[1], Julia Behnsen[2] and James D. Gardiner[3]

[1]Department of Natural Sciences, Manchester Metropolitan University, M1 5GD, UK
[2]Manchester X-ray Imaging Facility, University of Manchester, M13 9PL, UK
[3]Institute of Ageing and Chronic Disease, University of Liverpool, L7 8TX, UK

CAB, 0000-0002-6552-541X

The baculum is an enigmatic bone within the mammalian glans penis, and the driving forces behind its often bizarre shape have captivated evolutionary biologists for over a century. Hypotheses for the function of the baculum include aiding in intromission, stimulating females and assisting with prolonged mating. Previous attempts to test these hypotheses have focused on the gross size of the baculum and have failed to reach a consensus. We conducted three-dimensional imaging and apply a new method to quantify three-dimensional shape complexity in the carnivoran baculum. We show that socially monogamous species are evolving towards complex-shaped bacula, whereas group-living species are evolving towards simple bacula. Overall three-dimensional baculum shape complexity is not related to relative testes mass, but tip complexity is higher in induced ovulators and species engaging in prolonged copulation. Our study provides evidence of postcopulatory sexual selection pressures driving three-dimensional shape complexity in the carnivore baculum.

## 1. Introduction

The baculum bone is located within the glans penis of many modern mammals [1] and is extremely divergent in size and shape among closely related species [2]. Recent experimental work found a correlation between baculum morphology and reproductive success in rodents [3,4], yet the causative link between shape and reproductive output remains unclear. The baculum may aid intromission [5] or stimulate female partners [6]. Recently, a correlation between baculum robustness (a metric incorporating size and shape) and intromission duration was found in carnivores; taxa engaging in prolonged copulation are characterized by more robust bacula [7]. Yet within carnivores, some taxa possess divergent bacula shape while size remains relatively consistent across the group [8]. The evolution of baculum shape has not been fully explored.

This reflects difficulties in quantifying baculum shape. Within a rodent species, shape has been described using two-dimensional [4,9] and three-dimensional [10,11] geometric morphometrics (GMM). Such analyses require the identification of homologous landmarks and necessitate large numbers of sliding semi-landmarks given the often smooth, homogenous baculum surface. Applying GMM to diverse interspecific samples is problematic: bacula share few discrete landmarks, and the baculum may not in fact be homologous across mammals [1]. Investigating shape complexity is a viable alternative. Broadly speaking, complexity can be thought of as the number of 'parts' comprising a structure. In the context of biological shape variation, topographically more 'complex' shapes are those formed by combining a greater number of simple shape primitives (cubes, cylinders, spheres, tetrahedra, etc.) than less complex shapes. In practice, however, shape complexity has been quantified using a variety of methodological

approaches. For example, the two-dimensional shape complexity of water strider genitalia has been determined using outline-based elliptical Fourier analyses and dissection indices (perimeter:area ratio) and has been found to correlate with metrics of postcopulatory sexual selection [12]. Here, we apply the recently developed 'alpha shapes' methodology [13] to quantify three-dimensional shape complexity in the carnivore baculum and relate this metric to proxies for postcopulatory sexual selection pressures. By this definition, 'complexity' is calculated as the degree of refinement necessary in order for the volume of a fitted alpha shape to match that of an underlying baculum model (see Methods).

We hypothesize:

$H_1$. Baculum shape complexity will vary with the social system. In solitary families, polygyny and promiscuous mating is typical, while group-living families tend to be polygamous or polygynandrous [14]. Features of a 'complex' baculum (elaborate tips, urethral grooves) may aid sperm removal, female stimulation and prolonged intromission, behaviours expected under strong postcopulatory sexual selection. Social groups under high levels of postcopulatory sexual selection should possess complex bacula. Monogamous groups are expected to be under lower levels of postcopulatory sexual selection and possess less complex bacula.

$H_2$. Baculum shape complexity will be positively correlated with relative testes mass. Expenditure on spermatogenesis is expected to increase with postcopulatory sperm competition risk in order to increase a male's fertilization success per mating [15]. Relative testes mass is a commonly used proxy for the strength of postcopulatory sexual selection in birds [16–18] and mammals [19–22]. Given that features of 'complex' bacula may function in the delivery/removal of semen, we expect a relationship between three-dimensional shape complexity and relative testes size.

$H_3$. Baculum shape complexity will increase distally, and tip shape will be more complex in taxa characterized by induced ovulation and/or prolonged copulation. The baculum tip more directly interacts with the upper female reproductive tract and location of sperm deposition and may be more tightly correlated to reproductive strategies.

## 2. Methods

Bacula ($n = 82$) were loaned from osteological collections of the National Museum of Scotland, Edinburgh (NMS), the Natural History Museum, London (NHM) and the Smithsonian National Museum of Natural History (NMNH) (see electronic supplementary material, S1 for full list of specimens). Species sampled represent 12 out of 15 extant carnivore families and achieve 51% coverage of the total branch lengths of the consensus carnivore phylogeny (excluding those taxa for which the baculum is absent). Specimens with obvious pathologies or belonging to clearly subadult skeletons were excluded from the analysis. Individuals with associated life-history data (age and body mass) were preferred wherever possible. NHM specimens were micro CT scanned at the museum's Imaging and Analysis Centre, using a Nikon Metrology HMXST 225 scanner. NMS individuals were scanned at the Manchester X-Ray Imaging Facility in a 320/225 Nikon X-Tek Custom Bay. Resolution ranged from 7 μm (felids) to 60 μm (pinnipeds), with an average of 31 μm. Further details of the scanning parameters are provided in a previous paper [7].

The 32-bit .raw CT files were imported into Fiji, converted to binary data using an automated threshold based on the histogram of greyscale values, and exported as 8-bit .raw files. The data was subsequently imported in MATLAB R2017a (The MathWorks Inc., Natick, MA, USA) for alpha shapes analysis. The alpha shape methodology is provided in detail by Gardiner et al. [13]. Briefly, internal cavities present within the bacula were filled and CT data then converted from voxels to a three-dimensional point cloud. Point clouds were then downsampled such that every specimen was represented by 100 000 points and scaled in order to remove the effect of size from the analysis. Alpha shapes were fitted to the data using the MATLAB 'alphavol' function written by Jonas Lundgren (http://www.mathworks.co.uk/matlabcentral/fileexchange/28851-alpha-shapes). In addition, a small number ($n = 9$) of photogrammetric surface models, derived from the NMNH, supplemented the microCT-derived data. The alpha shapes protocol (above) was modified, such that each surface mesh was filled internally until each model was represented by a point cloud of 100 000 random vertices.

In the case of the baculum, alpha complexity can intuitively be thought of as the degree to which the structure departs from a simple rod shape. The protocol begins by fitting a very coarse convex hull object around the three-dimensional data. This hull inevitably overestimates the volume of the underlying object. The fitted shape is then progressively refined via the parameter 'alpha' to include both convex and concave regions, and the shape's volume eventually intersects that of the underlying baculum (figure 1). Optimal alpha complexity for each specimen was defined as the degree of refinement necessary in order for the volume of the alpha shape mesh to match that of the CT data (in which holes have been filled) or the original photogrammetry model. An optimization approach was therefore used, applying the 'fminsearch' function of MATLAB's optimization toolbox to calculate a single value for optimal alpha for each specimen. Here, three-dimensional shape complexity is reported as 1/refinement coefficient (as defined by Gardiner et al. [13]), such that outwardly more 'complex' structures are characterized by higher values of optimal alpha complexity. Regional variation in baculum complexity was quantified by fitting alpha shapes to three subregions of equal proportion along the long axis of the point cloud.

As associated body masses were rarely available for individual specimens, body masses were predominantly assigned from the literature ($n = 81$) (electronic supplementary material, S1). Likewise, data on the social system ($n = 82$) were also assembled from existing publications (electronic supplementary material, S1). Mating systems are likely to more closely reflect the strength of sexual selection under which genital structures are evolving. However, as noted elsewhere, the occurrence of polygyny in most carnivores is poorly documented [23], and the social system is therefore used as a proxy as per previous studies [23–25]. When this data was unavailable for a given species, the social system was assigned as per a sister taxon within the same genus ($n = 10$). Testes mass ($n = 55$) was also sourced from the literature, preferably accompanied by body mass . In some instances, this data included both testes plus epididymis mass and therefore represents the upper bound of likely total testes size. Additionally, in the absence of alternative data, testes volume was occasionally taken as representative of testes mass, under the assumption of an average testes density of $0.81 \text{ kg m}^{-13}$ (see electronic supplementary material, S1). Species for which testes mass was unavailable are listed as 'NA' in electronic supplementary material, S1.

Ovulation strategy ('spontaneous' versus 'induced') was likewise collated from the literature and is here considered as a binary trait. However, ovulatory style is notoriously difficult to ascertain, even in captivity, and species are increasingly recognized to be capable of either strategy depending upon physical/

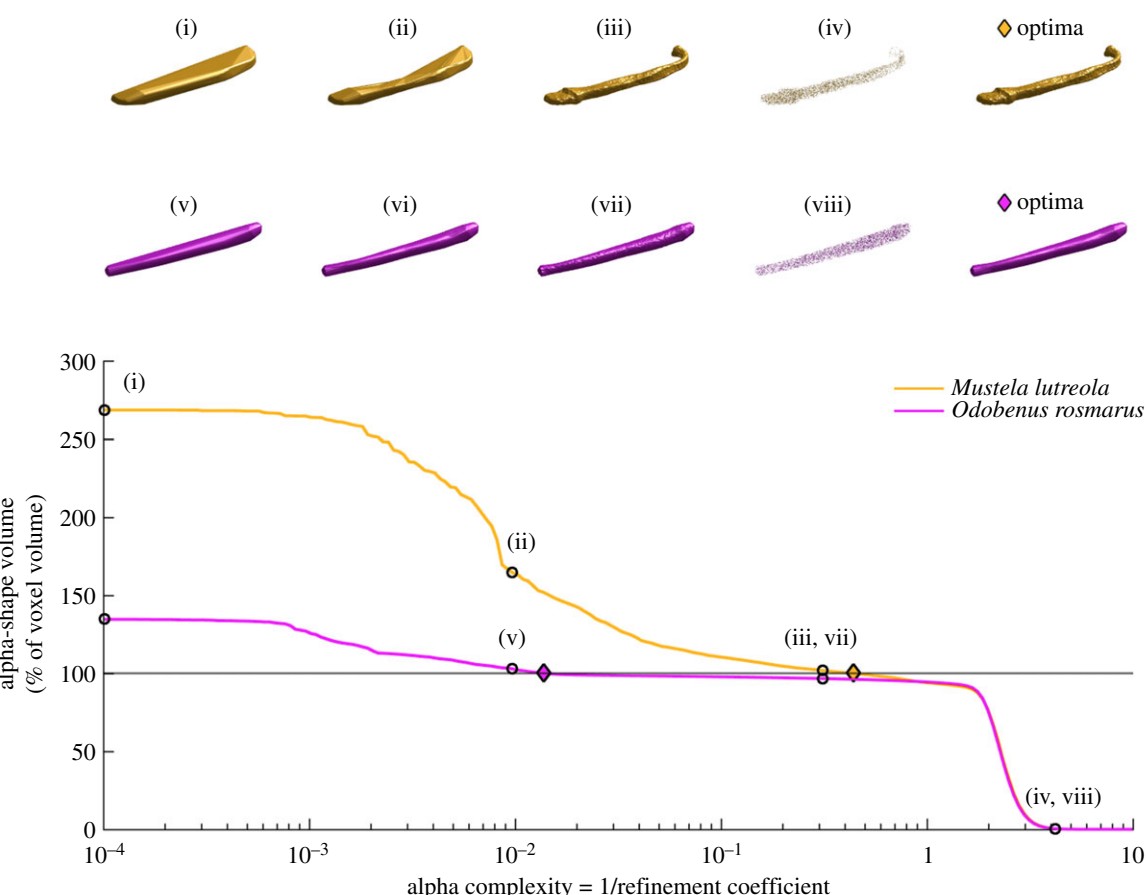

**Figure 1.** The alpha shape protocol for estimation of three-dimensional shape complexity. As the alpha complexity is increased (left to right on the *x*-axis), the alpha shape fitted to the underlying CT data becomes increasingly refined. Alpha complexity is defined as the inverse of Gardiner *et al.*'s [13] refinement coefficient, such that increasing shape complexity is represented by increasing values. The alpha complexity necessary in order for the volume of the alpha shape to match that of the CT dataset (100% on the *y*-axis) is reported as the 'optimal' alpha complexity. An outwardly 'simple' baculum (walrus, *Odobenus rosmarus*) rapidly converges on optimal alpha complexity at relatively coarse level of refinement coefficient (purple diamond; inset v–viii). By contrast, a seemingly more 'complex' baculum (European mink, *Mustela lutreola*) requires a much more refined alpha shape fit (yellow diamond; inset i–iv) in order for the model volumes to converge. Beyond this point, at very high levels of alpha complexity, alpha shapes begin to break down and no longer form one contiguous shape. At this stage, alpha shape volumes become less than the volume of the original mesh. A more detailed description of the alpha shape methodology is available elsewhere [13]. (Online version in colour.)

psychosocial stimuli [26,27]. As per previous research [28], here, we consider an induced ovulator to be any species for which induced ovulation has been documented, while recognizing this trait is likely to fall along a continuum. For example, all felids are designated as induced ovulators here, despite modest-to-frequent spontaneous ovulation being recorded in an array of cats [29]. Further work is needed in order to quantify the relative frequency of ovulatory styles within carnivoran species and the factors underlying this physiology.

A consensus tree and tree block of 10 000 trees based on GenBank data was downloaded from the 10kTrees website [30] (electronic supplementary material, S2 and S3). The level of phylogenetic signal present in optimal alpha complexity was quantified using the 'phylosig' function of 'phytools' [31] to calculate Pagel's lambda ($\lambda$), based on the consensus carnivore phylogeny. Lambda values approaching 1 suggest traits follow a purely Brownian motion model of evolution, while lambda values of 0 suggest no correlation between species relative to the correlation expected under Brownian motion. Estimated values were compared to a null assumption of zero using a likelihood ratio test following a chi-squared distribution.

In order to determine how the social system may have influenced baculum shape complexity, an evolutionary model selection approach was used. The evolution of carnivore social system was inferred by stochastic character mapping using the 'make.simmap' function of 'phytools' across the posterior distribution of 10 000 trees. Evolutionary models were fitted using the R package 'OUwie' [32]. The simplest fitted model was single-rate Brownian motion (BM1), which does not allow for separate selective regimes on the basis of social system and assumes trait variance accumulates in proportion to time. Additionally, a multi-regime Brownian motion model (BMS) was fitted, which allows for the rate of stochastic motion $\sigma^2$ to vary between social systems. Ornstein–Uhlenbeck models were also considered; the single-peak Ornstein–Uhlenbeck (OU1) assumed a single phenotypic optimum $\theta$, while a multi-peak OU model (OUM) assumes a different $\theta$ for each regime while holding $\sigma^2$ and $\alpha$ (the rate at which traits evolve towards the optima) constant. Finally, a more complex OU model was fitted (OUMV) in which separate $\theta$ and $\sigma^2$ for each social system regime is permitted.

Results of all models were checked to ensure the eigenvalues of the Hessian were positive, indicating that the model parameters were reliably estimated [32]. When a negative value was identified, the results for that model and tree combination were discarded. Additionally, models returning nonsensical values for optimal alpha complexity (an order of magnitude greater than the most complex specimen, or an order of magnitude less than the least complex) were excluded from the analysis. The best overall model was selected in accordance with the Akaike information criterion corrected for small sample size (AICc; smaller values indicate greater support). Models with AICc values within ±2 of the best-fitting model were also considered to have significant support [33].

Phylogenetic regressions between baculum length and shape complexity and testes mass were conducted using the 'Continuous: Regression' module in BayesTraits v. 2 [34], which generates a posterior distribution of phylogenetic generalized least square (PGLS) regression models derived from the 10 000 trees sampled from a Bayesian posterior probability distribution from 10kTrees. BayesTraits was run within the BayesTraits Wrapper (btw) package in R by Randi Griffin (https://github.com/rgriff23/btw). Each model was run for 5 000 000 iterations, sampling every 500 iterations with a burn-in of 500 000. The scaling parameter lambda ($\lambda$) was simultaneously estimated to determine the extent to which the trait evolved as expected under the given topology. All data were $\log_{10}$-transformed prior to analysis. As per Brindle & Opie [35], each analysis was run three times and the chain with the median log marginal likelihood was chosen. This approach was also used when testing for correlated evolution (see below). The significance ($p$-value) of the regression models was calculated as the proportion of the posterior distribution of slope parameters ($\beta$) that crossed zero (the null model), as per the BayesTraits manual (http://www.evolution.rdg.ac.uk/BayesTraitsV3.0.1/Files/BayesTraitsV3.Manual.pdf) and Organ et al. [36].

In addition, we tested for coevolution between baculum length and shape complexity and normalized testes mass (testes mass/male body mass) using the 'Continuous: Random Walk MCMC' module of BayesTraits. This involved comparing the likelihood of posterior distribution of a dependent and independent analysis. The dependent analysis allowed for the evolution of one trait to be dependent upon the state of a second trait and should be the preferred model when coevolution has occurred. By contrast, the independent analysis assumes the evolution of one trait is independent of the other. As above, each analysis was run for 5 000 000 iterations, sampling every 500 iterations with a burn-in of 500 000. At the end of each analysis, log marginal likelihood was estimated using a stepping-stone sampler [37] across the posterior distribution of analyses, using 1000 stones run per 10 000 iterations of the Markov chain. As per previous studies [35,37], a beta distribution with $\alpha = 0.3$ and $\beta = 1$ was employed. The best-fit model of trait evolution was determined by the log(Bayes factor) or logBF, which was calculated as: logBF = 2(log[marginal likelihood(dependent model)] – log[marginal likelihood(independent model)]). On the logarithmic scale, a negative value of logBF indicates support for the independent model of evolution, i.e. traits are not co-evolving. Positive logBF values between 0 and 2 are taken as 'weak' evidence of coevolution, 2–5 as 'positive' evidence, and >5 as 'strong' evidence of trait coevolution [38].

When examining regional variation in baculum alpha complexity, phylogenetic t-tests were conducted using the 'Continuous: Regression' module in BayesTraits. The binary trait (ovulation strategy: spontaneous versus induced) was correlated against localized alpha complexity using the same MCMC procedure as described above (as per Opie et al. [39]). The relationship between localized alpha complexity and intromission time was likewise quantified using the 'Continuous: Regression'. Within-baculum variation in alpha complexity was quantified using a univariate repeated-measures ANOVA on $\log_{10}$-transformed data in the 'car' package [40] of R. As assumptions of data sphericity were violated (as assessed per a Mauchly test), Greenhouse-Geiser corrections were applied [41]. This test is not phylogenetically corrected however.

The extent to which the particularly small felid baculum remains 'functional' during copulation, as opposed to becoming a residual feature upon which selection no longer acts, has been the subject of debate (see Discussion). We therefore rerun the above analyses while excluding felids from the dataset, to explore the effect of their unusual morphology upon subsequent results (see electronic supplementary material, S4).

# 3. Results

We reconstructed the evolution of optimal alpha complexity by mapping the trait onto a composite carnivore phylogeny sourced from the 10kTrees project (figure 2) and quantified the strength of phylogenetic signal present in the data in terms of Pagel's lambda ($\lambda$). A strong statistically significant phylogenetic signal was identified in optimal alpha shape complexity ($\lambda = 0.89$, logL = −13.08, $p < 0.001$; figure 2). We found ursids and pinnipeds to possess bacula with low shape complexity, while cats, herpestids and musteloid mustelids were characterized by high alpha complexity. When members of the Felidae family were removed, this phylogenetic signal remained highly significant ($\lambda = 0.90$, logL = −13.4, $p<0.001$).

## (a) Evolutionary model selection

We investigated whether carnivore social systems are characterized by different evolutionary optimal values of alpha complexity using an evolutionary model selection approach. Comparison between AICc scores for all evolutionary models fitted to alpha shape complexity values (figure 3; table 1) shows strong support (i.e. lower AICc scores) for OUMV and BMS models, in which either/or $\theta$ (the phenotypic optima) and $\sigma^2$ (the rate of stochastic motion around the optima) are allowed to vary between social systems. Models assuming a single evolutionary regime (BM1 and OU1) are comparatively poorly supported, as is the multi-peak OUM model in which $\theta$ varies but $\sigma^2$ remains fixed across regimes.

Of 10 000 analysed stochastic character maps, 56% are best-fitted by an OUMV model and 44% by BMS. In 83% of character maps, a second model (BMS or OUMV) fell within ±2 AICc values of the best-fitting model. The results clearly illustrate that the evolution of three-dimensional shape complexity in the baculum is characterized by differing trait optima ($\theta$) between social systems, and this is achieved primarily through differences in the rate of stochastic fluctuation ($\sigma^2$).

Regardless of the best-fitting model chosen (BMS or OUMV), the estimated parameters describe a similar pattern (figure 4). The baculum of group-living carnivores (defined as species in which 'several breeding females share a common range and forage or sleep together' [14]) is evolving towards a very simplified shape, yet with a high intensity of random fluctuations around this optimum. By contrast, the baculum of socially monogamous species (species in which 'a single breeding female and single breeding male share a common range or territory and associate with each other for more than one breeding season, with or without non-breeding offspring' [14]) is evolving towards a highly complex shape with less intense fluctuations around the trait optima. Model parameters for solitary carnivores (species in which 'breeding females forage independently in individual home ranges and encounter males only during mating' [14]) are intermediate, falling between the two other social categories. When this analysis is repeated with the exclusion of felids, all patterns described above remain identical (see electronic supplementary material, S4).

# 4. Baculum shape complexity in relation to testes mass

We investigated whether baculum length and shape complexity correlated with relative testes mass, a metric often

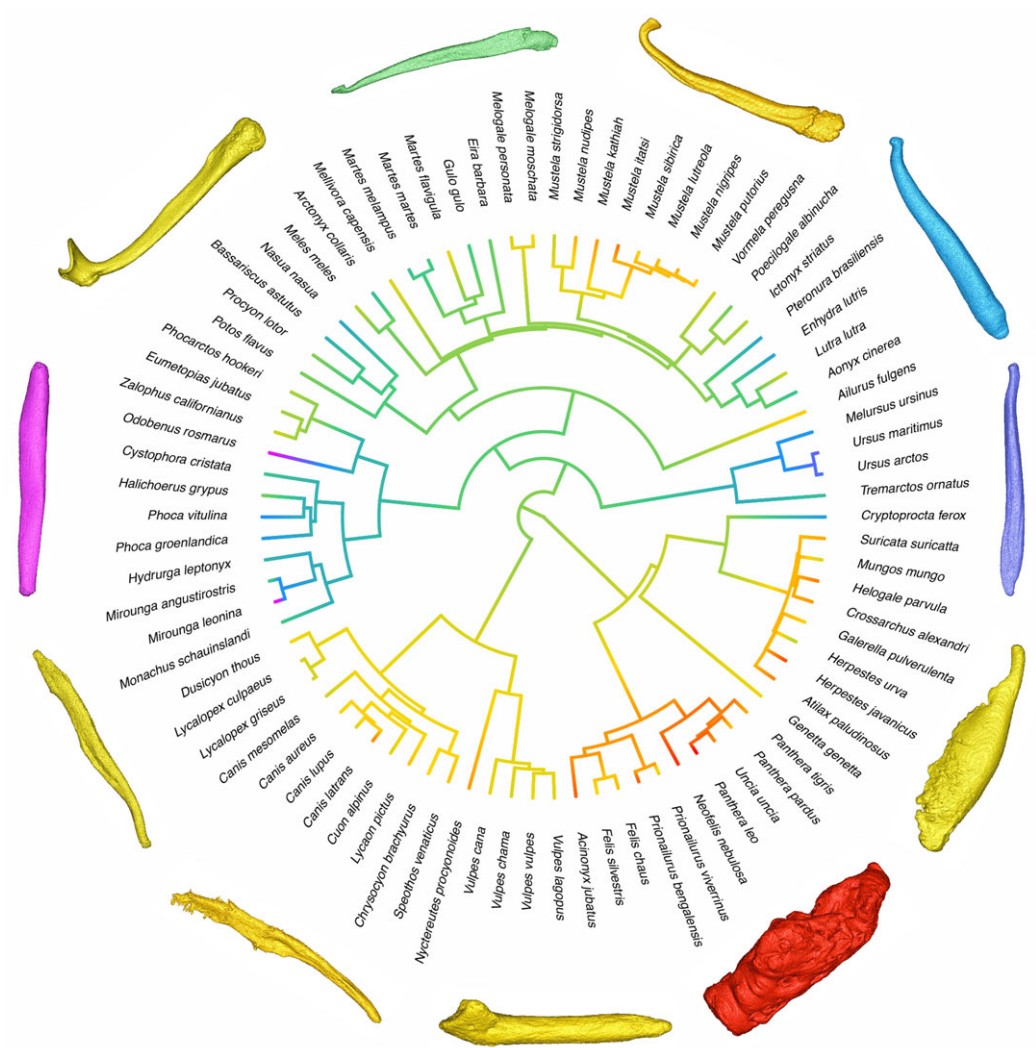

**Figure 2.** Ancestral state reconstruction of $\log_{10}$ baculum complexity across Carnivora. Consensus phylogeny sourced from 10kTrees project (https://10ktrees.nunn-lab. org/index.html). Trait evolution was reconstructed using the 'phytools' function 'fastAnc' and plotted using 'contMap'. Cool colours = low alpha shape complexity; hot colours = high alpha shape complexity. Dorsal surface of three-dimensional models faces outwards from the tree root, distal tips point in an anti-clockwise direction. Clockwise from top: *Mustela lutreola*, *Pteronura brasiliensis*, *Ursus maritimus*, *Mungos mungos*, *Panthera leo*, *Felis silvestris*, *Speothus venaticus*, *Canis lupus*, *Mirounga leonina*, *Mellivora capensis*, *Eira barbara*. Example species for three-dimensional surfaces were selected as being representative of their closely related taxa. (Online version in colour.)

taken as a proxy for the strength of postcopulatory sexual selection. The relationship between baculum size and shape and testes mass was assessed using separate multiple regressions with male body mass added as a covariate to account for allometric effects [42]. The posterior distribution of regression parameters is illustrated in electronic supplementary material, S5 and average parameter values are provided in table 1 and 2. Baculum length was found to be significantly, albeit very weakly, positively correlated with residual testes mass ($p = 0.04$; table 2), while baculum complexity was not ($p > 0.05$).

In addition, we tested for coevolution between the carnivoran baculum and testes size by comparing the posterior distribution of models in which (i) baculum length and normalized testes mass evolve independently of one another, and (ii) baculum complexity and normalized testes mass evolve independently of one another, against models in which the two traits co-evolve. In this second scenario, the probability of a change in one trait depends upon the value of the second trait. In line with the above PGLS regression results, we found no support for the 'dependent' model, and neither

baculum length (logBF = −0.015) nor complexity (logBF = 0.097) was found to co-evolve with normalized testes mass.

In some instances ($n = 5$), testes mass from the literature also included epididymis mass. The analysis was repeated excluding these individuals (electronic supplementary material, S4), but all trends reported above remain the same.

## 5. Baculum shape complexity in relation to reproductive strategies

We investigated within-baculum variation in shape complexity, quantifying alpha complexity in the proximal attachment site, midshaft and distal tip of the carnivore baculum. A univariate within-subject ANOVA with Greenhouse-Geisser correction for non-sphericity identified significant regional variation in alpha complexity ($\varepsilon = 0.82$, $F_{26.4, 283.4}$, $p = <0.001$). Pairwise comparison of means found both the proximal ($p < 0.001$) and distal ($p < 0.001$) ends to be significantly more complex than the midshaft, yet the proximal

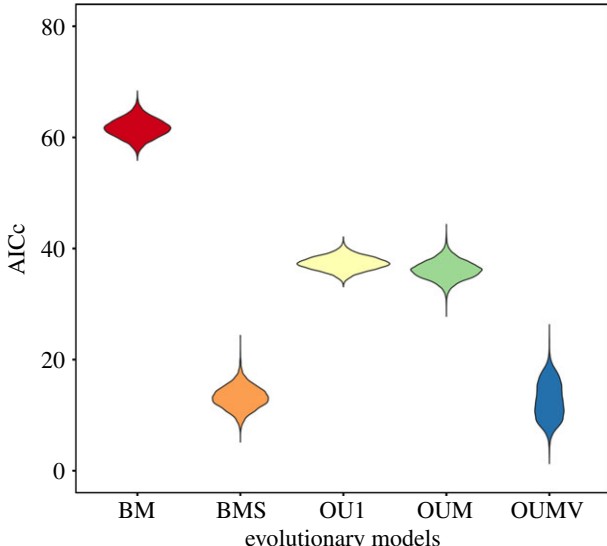

**Figure 3.** Evolutionary models scored according to the Akaike information criterion corrected for small sample size (AICc). Results from 10 000 simulations, using the OUwie package of R. Smaller values of AICc indicate greater support for the model. BM, Brownian motion; BMS, multi-regime Brownian motion model; OU1, single-peak Ornstein–Uhlenbeck model; OUM, multi-peak Ornstein–Uhlenbeck model; OUMV, multi-peak Ornstein–hlenbeck model with variable $\sigma^2$ between social systems. (Online version in colour.)

attachment and distal tip were not significantly different from each other ($p = 0.75$).

Using a phylogenetically corrected generalized least-squares approach, we examined whether regional shape complexity of the baculum was correlated to such carnivoran reproductive strategies (see electronic supplementary material, S1). We found neither whole-bone complexity, midshaft nor proximal attachment complexity were correlated to intromission period ($p > 0.05$). However, baculum tip complexity was significantly, albeit weakly, correlated to intromission duration ($n = 54, \lambda = 0.83, r^2 = 0.11, p = 0.02$), such that carnivores copulating for extended durations possess more complex distal tip elaborations. Likewise, tip complexity was weakly correlated to ovulation mechanism ($n = 72, \lambda = 0.68, r^2 = 0.04, p = 0.039$), where taxa experiencing induced ovulation are characterized by more complex distal tip shapes than spontaneous ovulators (see electronic supplementary material, S5). This not the case for baculum whole-bone alpha complexity, nor midshaft or proximal attachment site complexity ($p > 0.05$; see electronic supplementary material, S5).

All trends reported above remain the same when felids are excluded from the dataset, with the exception of the relationship between tip complexity and intromission duration which becomes marginally insignificant ($n = 47, \lambda = 0.89, r^2 = 0.06, p = 0.053$).

## 6. Discussion

### (a) Three-dimensional alpha shape complexity of the carnivore baculum

We found values for optimal alpha complexity to closely reflect previous qualitative descriptions of baculum complexity. Bear and pinniped bacula have been described as being mostly straight, rod-like, with slight distal tapering [43], while the elaborate distal tips and curvature of mustelid bacula have been singled-out as 'complex' in the literature [8]. Alpha shapes also successfully highlight more subtle taxonomic trends, such as the comparatively complex baculum of otariids versus phocids and the walrus, and the notably simple baculum of otters when compared to other mustelids.

Felids are a notable exception; however, their baculum having previously been described as 'rudimentary' or 'residual' [44] and often analysed separately or excluded entirely [28,45]. While undoubtedly diminutive in length relative to body size, our scale-invariant alpha complexity approach identifies felid baculum as highly complex. This may be attributable to the unusual structure of the proximal baculum region. Although simple and 'rod-like' distally, the proximal portion of the cat baculum is often characterized by paired, deep depressions, divided by a dorsoventrally aligned septum corresponding to the terminal ends of the corpora cavernosa [44]. These depressions can be extensively developed and probably contribute to the seemingly 'complex' structure of the felid baculum. Furthermore, the connection between the baculum and corpus cavernosum is characterized by the insertion of collagen fibres originating from the tunica albuginea deep into the bone. Elsewhere in the vertebrate body, the interfacial roughness of such entheses is found to be scale-invariant [46], and the degree to which such features are considered 'complex' by alpha shapes will vary with respect to the absolute size of the baculum. Such scaling challenges provide scope for further methodological advancements. Regardless of whether such 'complex' features are indeed 'functional', our sensitivity analysis finds the inclusion/exclusion of felid bacula does little to affect the results of our downstream analyses.

### (b) Baculum shape complexity varies between carnivore social systems

Our initial hypothesis that baculum shape complexity would vary between carnivore social systems was supported. Yet, contra to our expectations, 'socially monogamous' species are found to possess high values for optimal baculum complexity. This is probably driven by socially monogamous carnivores being comprised mostly of canids. The assumption that social systems adequately represent the strength of postcopulatory sexual selection is a limitation of this approach. Social monogamy is not equivalent to genetic monogamy, and extra-pair paternities are common among canids [47]. While polygynandrous/communal breeding has been documented in *Lycaon pictus* [48], for example, here the species is considered to be socially monogamous. Features of the canid baculum acting to increase alpha complexity (a deep urethral groove, scarring associated with the bulbus glandis attachment) are associated with the unique canid copulatory tie, itself thought to function in a mate-guarding capacity within a polygamous system [49,50].

Likewise, here species considered to be 'group-living' on the basis of Lukas & Clutton-Brock's [14] definition also include some harem-living pinniped genera such as *Odobenus, Halichoerus* and *Mirounga*. In such species, precopulatory sexual selection is high, and a small number of successful males secure access to breeding females. In this particular scenario, 'group-living' is therefore likely to be associated with reduced sperm competition [22]. It is perhaps unsurprising, therefore, that such taxa are characterized by comparative simple

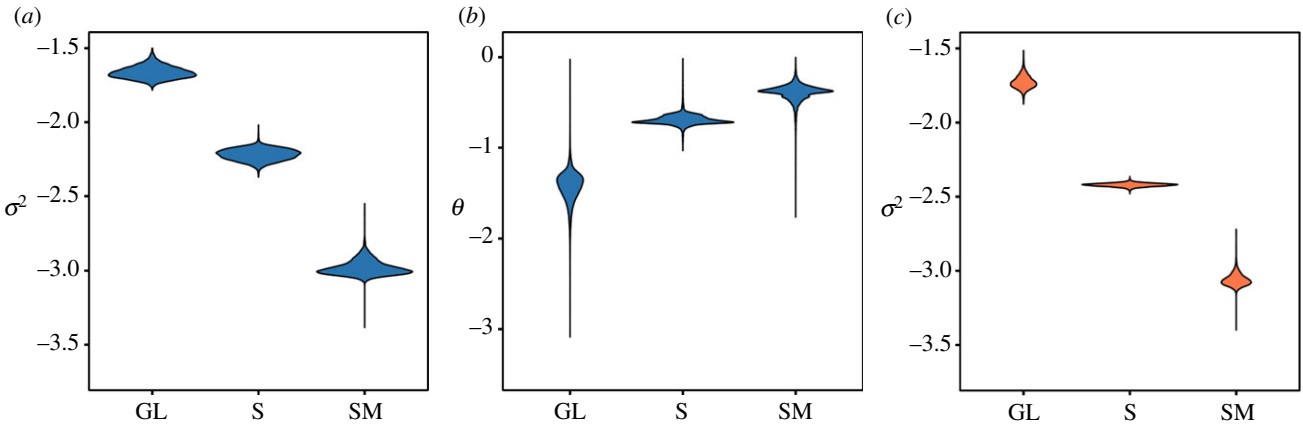

**Figure 4.** Evolutionary model parameters. Results from 10 000 simulations using the OUwie package of R. Sigma-squared ($\sigma^2$), nondirectional rate of evolution; theta ($\theta$), trait optima. (*a,b*) Parameters estimated under OUMV model of evolution; (*c*) parameter estimated under BMS model of evolution. GL, group-living; S, solitary; SM, social monogamy. (Online version in colour.)

**Table 1.** Comparing evolutionary model fits for baculum complexity. AICc, mean Akaike information criterion corrected for small sample size; $\Delta$AICc, the difference between the best model (smallest AICc) and each model; AICcW, Akaike weights, calculated as the relative likelihood of the model divided by sum of likelihoods.

| model | AICc | $\Delta$AICc | AICcW |
|-------|------|--------------|-------|
| BM | 61.8 | 49.02 | 0.00 |
| BMS | 13.2 | 0.52 | 0.44 |
| OU1 | 37.3 | 24.67 | 0.00 |
| OUM | 36.2 | 23.50 | 0.00 |
| OUMV | 12.7 | 0.00 | 0.56 |

bacula (figure 1). In the absence of improved data on carnivore mating systems (see Methods), we therefore proceeded by using relative testes mass as a potentially more refined proxy for the strength of postcopulatory sexual selection.

## (c) Baculum shape complexity does not correlate with relative testes mass

Baculum shape complexity was found to neither co-evolve nor correlate with relative testes size. This agrees with results found elsewhere across the Carnivora order [51] and in specific clades (pinnipeds[22]), yet counter to a recent MCMC-based analysis on carnivorans [35]. Our results suggest postcopulatory sexual selection (as determined by relative testes size) acts to promote the lengthening of the baculum in carnivores. By contrast, alpha complexity of the baculum was not significantly correlated with residual testes mass (table 2), suggesting that three-dimensional shape complexity as estimated across the entire baculum is not evolving in response to postcopulatory sexual selection pressures.

We therefore reject our original hypothesis that baculum complexity should increase with relative testes size. This is perhaps surprising, as alpha complexity does appear to be correlated with the appearance of many features (urethral grooves, apical hooks etc) that may be adaptive under a regime of strong postcopulatory sexual selection (figure 1).

Our findings may, in part, reflect shortcomings associated with the use of literature values for testes mass. Such data are often sparsely reported, sensitive to variation in measurement protocol [52] and will, particularly in carnivores, be subject to seasonal change. We do, however, find relative testes mass to be weakly correlated to baculum length as per previous studies [22,51], suggesting the testes data included here are comparable to those collected elsewhere.

Alternatively, the lack of correlation between shape complexity and relative testes mass may be attributed to the 'multitrait' nature of genitalia. As discussed elsewhere [12], male intromittent organs must fulfil multiple tasks (potentially including intromission, stimulation, ejaculation and sperm removal, in addition to waste excretion). Some degree of modularity in the shape of the baculum is therefore expected, as different regions of the element are probably subject to contrasting selective pressures, and each of these subregions will contribute to the overall calculation of complexity. While a single metric of shape complexity calculated across the baculum may therefore reflect the 'multipurpose' nature of carnivore genitalia, the gross nature of alpha complexity may also cloud any regional patterns in trait divergence.

## 7. Baculum tip complexity correlates with carnivore reproductive strategies

We identified significant differences in alpha complexity between the tip/base regions of the baculum and the midshaft. The expectation that regional variation in complexity will exist within the baculum is therefore supported. The baculum is located within the glans tissue of the penis and is anchored at the proximal end to the paired corpora cavernosa via a layer of fibrocartilage [53], resulting in rugose paired attachment sites. The morphology of the proximal attachment site does differ between carnivore families however. In mustelids, for example, the proximal baculum is mediolaterally compressed and characterized by low relief nodules, while the pinniped proximal attachment site remains sub-elliptical in cross-section with fibrous surface texture. As previously mentioned, the proximate felid baculum possesses paired 'depressions' divided by a dorsoventral septum. The proximal attachment site of the baculum has been hypothesized to play a role in increasing

**Table 2.** Average regression parameters derived from a posterior distribution of multiple PGLS analyses conducted in BayesTraits in the form 'baculum parameter ~testes mass + body mass'. $\lambda$, lambda (indicates degree to which traits are evolving as predicted given the underlying tree topology); $\alpha$, intercept; $r^2$, coefficient of determination; $\beta$, slope; $p$, significance of the model; CI, 95% confidence intervals. Significant ($p < 0.05$) values indicated by an asterisk. Baculum length in mm. Body mass in kg. Testes mass in g. All data is $\log_{10}$-transformed prior to analysis.

| trait | n | $\lambda$ | $\alpha \pm$ SE | $r^2$ | predictor | $\beta \pm$ SE | $p$ | $\beta$ CI |
|---|---|---|---|---|---|---|---|---|
| baculum length | 55 | 0.93 | $1.30 \pm 0.15$ | 0.43 | testes mass | $0.15 \pm 0.09$ | 0.042* | 0.01–0.30 |
| | | | | | body mass | $0.16 \pm 0.08$ | 0.032* | 0.02–0.30 |
| baculum complexity | 55 | 0.95 | $-0.60 \pm 0.24$ | 0.06 | testes mass | $0.09 \pm 0.13$ | 0.191 | −0.11–0.33 |
| | | | | | body mass | $-0.21 \pm 0.12$ | 0.039* | −0.42–0.02 |

penile hydrostatic pressure through a 'plunging' action, transferring forces from the distal glans to the tensile walls of the corpus cavernosum [53]. The degree to which the complexity in proximal baculum shape noted here is reflected across carnivores within the associated soft tissue attachments and in its biomechanical performance *in situ* remains to be tested.

By contrast, the baculum shaft of most carnivores approximates a simple cylinder with some degree of curvature along the long axis, and this simplicity is reflected in alpha complexity values calculated herein. As an exception, the ventral surface of canid bacula is characterized by a well-developed urethral groove, extending to the distal tip. Less pronounced ventral grooves are also apparent in some mustelid and pinniped taxa. In canids, the dorsal surface of the baculum slightly proximal to midshaft may also be raised with a rugose bone texture, reflecting the location of the bulbus glandis erectile tissue associated with the prolonged copulatory tie.

Here we found that neither whole-bone, proximal nor midshaft complexity correlated to intromission duration (electronic supplementary material table S2). Therefore, while the presence of midshaft complexity (such as deep urethral grooves and bulbus glandis scaring) is characteristic of some prolonged maters such as canids, this pattern does not extend to other families engaging in high duration intromission, including mustelids. However, shape complexity within carnivore bacula increases at the distal tip. The distal end of the baculum terminates within the glans tissue, and in some taxa is characterized by hooks, 'scoops', bifurcations and other elaborate condyles. In other taxa, however, the baculum simply ends as a blunt rounded terminus. Here, we identified a weak positive correlation between baculum tip complexity and intromission duration. This mirrors the previous results of Brassey *et al.* [7]'s biomechanical analysis, in which simulated stress values induced under bending applied in the ventral direction were also found to negatively correlate with intromission duration. In addition to increasing complexity in the present analysis, the distal hook probably acts to mitigate stress under dorsoventral bending when load is hypothetically applied at the tip. Likewise, we also find induced ovulators to be characterized by more complex bacula tips.

Although significant, the weak positive correlations identified above suggest our analysis may still be failing to capture some element of genital complexity and/or copulatory behaviour in mammalian carnivores. The data presented here are derived from museum osteological collections, and in all instances, the baculum is represented by a single ossified element. In reality, however, an additional cartilaginous tip has been documented in canids [54,55], and its broader phylogenetic distribution has not been systematically investigated. This element is currently absent from most museum specimens and hence from our analysis. Likewise, our analysis implicitly assumes baculum complexity to be an accurate proxy for penile shape complexity. Unlike the previous application of shape complexity metrics of invertebrate genitalia, however [12], here the baculum lies surrounded by glandular tissue, the extend of which appears to vary considerably within Carnivora. The mustelid baculum, for example, is covered by a thin sheath of soft tissue, and the morphology of the baculum and distal erect penis are expected to be closely aligned. By contrast, the baculum of most feliformes is surrounded by a thick layer of soft tissue, and the shape complexity of the erect distal glans may bear little resemblance to that of the os penis. Future research efforts focusing on quantifying the morphology of erect penile soft tissues, using recently developed inflation and fixation techniques [56,57] will go some way to redressing these gaps in our knowledge.

Likewise, a broader critique of diverse interspecific morphological studies is the frequent failure to capture potential within-species variation. While the use of the baculum as a species diagnostic character suggests comparatively little intraspecific variation is present, some degree of static allometry has been documented in adult bacula of polecats [58], badgers [59], stone martens [60] and Eurasian wolves [61]. Similarly, experimental evidence increasingly points towards metrics other than baculum length as being the target for post-copulatory sexual selection within adult rodents [3,4]. The future deployment of alpha shapes to quantify intraspecific three-dimensional baculum complexity will undoubtedly prove essential in further unravelling within- versus between-species shape variation and its associated evolutionary drivers.

In support of our original hypothesis, we find that three-dimensional shape complexity in the baculum does increase distally relative to the midshaft. This was expected, on the grounds that the distal baculum more directly interacts with the female reproductive tract. However, very little is actually known of the three-dimensional morphology of female genitalia within carnivores, with the exception of some pinniped taxa [57]. Yet increasingly, the female tract is recognized as a dynamic and complex environment. The recent application of novel shape quantification tools to invertebrates has, for the first time, found female genital traits can show faster rates of evolutionary divergence than the male traits with which they interact [62]. Likewise, modern three-dimensional

imaging has revealed previously undocumented complexity in female genitals [63], and experimental manipulation studies [57,64] are exposing heretofore hidden internal dynamics of copulation. Such methodological advances are revolutionizing our understanding of the interaction between male and female genitalia, and future research should aim to contextualize the evolution of the carnivoran baculum within an improved understanding of the three-dimensional geometry of the corresponding female tract.

## 8. Conclusion

The mammalian baculum is characterized by astonishing morphological diversity. Yet remarkably, this widespread appendicular bone had previously eluded a rigorous quantitative analysis of shape across a broad range of taxa. In part, this has reflected methodological challenges associated with the application of shape variation metrics to such disparate structures. For the first time, here we have quantified three-dimensional shape complexity in the carnivoran baculum, using the recently developed 'alpha complexity' methodology.

Three-dimensional shape complexity in the carnivoran baculum is strongly distributed along phylogenetic lines. Socially monogamous taxa such as canids possess bacula distinguished by features such as deep urethral grooves and rugose attachment sites, and are accordingly evolving towards a higher value of optimal shape complexity. By contrast, group-living species such as pinnipeds possess simple rod-like bacula found to be evolving towards low values of optimal alpha complexity. When calculated across the baculum as a whole, three-dimensional shape complexity does not correlate with relative testes mass. However, a regional analysis finds increased tip complexity to be associated with prolonged intromission duration and stimulated ovulation. Taken together, our results provide evidence for postcopulatory sexual selection acting to drive the evolution of shape complexity in at least some regions of the carnivore baculum and provide a valuable new framework within which to analyses baculum shape. Future research will focus on the biomechanical function of specific regions of the baculum and will benefit from an improved understanding of the surrounding penile soft tissues and the geometry of the corresponding female reproductive tract.

Data accessibility. Shape complexity data and associated life-history traits are provided in the electronic supplementary material.

Authors' contributions. C.A.B. conceived of the study, designed and coordinated the study, carried out the statistical analyses and drafted the manuscript. J.B. participated in data collection and analysis, and critically revised the manuscript. J.D.G. participated in the design of the study, data analysis and critically revised the manuscript. All authors gave final approval for publication and agree to be held accountable for the work performed therein.

Competing interests. We declare we have no competing interests.

Funding. C.A.B. is funded by a BBSRC Future Leader Fellowship (BB/N01957/2). Manchester X-ray Imaging Facility is funded in part by the EPSRC (grant nos. EP/F007906/1, EP/F001452/1 and EP/102249X/1).

Acknowledgements. The authors would like to acknowledge Dr Andrew Kitchener and Georg Hantke (National Museum of Scotland). We thank Anna Herdina and two anonymous reviewers for their valuable feedback and role in significantly improving the manuscript.

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
