## [Reviewer comments · Proceedings of the Royal Society B: Biological Sciences]

Review History

RSPB-2020-0776.R0 (Original submission)

Review form: Reviewer 1

Recommendation

Major revision is needed (please make suggestions in comments)

Scientific importance: Is the manuscript an original and important contribution to its field?

Good

General interest: Is the paper of sufficient general interest?

Excellent

Quality of the paper: Is the overall quality of the paper suitable?

Good

Is the length of the paper justified?

Yes

Should the paper be seen by a specialist statistical reviewer?

No

Do you have any concerns about statistical analyses in this paper? If so, please specify them explicitly in your report.

No

It is a condition of publication that authors make their supporting data, code and materials available - either as supplementary material or hosted in an external repository. Please rate, if applicable, the supporting data on the following criteria.

Is it accessible?

Yes

Is it clear?

Yes

Is it adequate?

Yes

Do you have any ethical concerns with this paper?

No

Comments to the Author

Brassey et al. Postcopulatory sexual selection drives the evolution of shape complexity in the carnivoran baculum. This is an interesting paper and topic. The paper is very well written, and the approaches seem reasonable. The authors are very candid about difficulties of the study, including how to score induced vs. spontaneous ovulators. They are also very candid in presenting results that were contrary to some of their predictions (this should be commended).

In general, however, I think this dataset needs to be expanded in order to validate the conclusions.

1) I recognize that defining complexity is difficult, but some biological meaning is missing here. Under their definition, felid bacula are considered "most complex". However, this is driven by the fact that felid bacula are full of vacuities (as they discuss). These vacuities, in combination with the extremely small size and overall amorphous lumpiness of felid bacula suggest they are non-functional, perhaps even on an evolutionary trajectory of being lost. At the very least, felid bacula are probably not subject to the same selective pressures that other carnivoran bacula experience. Complexity can arise for a diverse number of methodological reasons, without biological function.

2) I am concerned about the small number of species analyzed. Let's take what appears to be two extremes on the complexity scale: felids which are most complex, and pinnipeds which are least complex. All felids are scored as "S" social system, and all pinnipeds are scored "GL" (their supplementary material). A major conclusion is that S species have more complex bacula and GL species are less complex, but this could be entirely driven by just these two groups. What is most concerning is that there are plenty of pinniped species with much more complex bacula (i.e., *Zalophus*) that would be scored GL and thus reduce the strength of the correlation to social system. The larger issue is that there are many more species that should be readily available from natural history museums (raccoons, skunks, etc.). I recommend expanding the number of species to validate the inferences.

3) At the very least, the authors need to explore if/how their results change after subsampling (i.e., jackknifing) their data. For example, felids are deemed "complex" and also induced ovulators. They represent a disproportionate number of species in the dataset, so are they driving the correlation between tip complexity and induced ovulation? Or between social system and complexity? Do the results hold after removing felids? What about removing other groups? The main question is how stable are these results as different groups are removed from the dataset?

4) It is difficult to interpret patterns of interspecific divergence without replicate

individuals within a species. The current dataset has sampled one baculum per species, would be good to include multiple representatives per species whenever possible.

Minor:

- 1) For the phylogenetic regressions between complexity, length, and testes mass – doesn't it make more sense to use testes mass residuals?
- 2) Line 334 – testing differences between sections of the baculum... were these phylogenetically controlled tests?

Review form: Reviewer 2

Recommendation

Accept with minor revision (please list in comments)

Scientific importance: Is the manuscript an original and important contribution to its field?

Excellent

General interest: Is the paper of sufficient general interest?

Good

Quality of the paper: Is the overall quality of the paper suitable?

Good

Is the length of the paper justified?

Yes

Should the paper be seen by a specialist statistical reviewer?

No

Do you have any concerns about statistical analyses in this paper? If so, please specify them explicitly in your report.

No

It is a condition of publication that authors make their supporting data, code and materials available - either as supplementary material or hosted in an external repository. Please rate, if applicable, the supporting data on the following criteria.

Is it accessible?

Yes

Is it clear?

Yes

Is it adequate?

Yes

Do you have any ethical concerns with this paper?

No

Comments to the Author

The authors use a recently developed measure of shape complexity to examine baculum shapes in carnivores. They find a strong phylogenetic signal in baculum complexity across carnivores. Different social systems showed evidence of different 3D shape trait optima (θ). Two kinds of

models best fit the data (Multi-regime Brownian motion model and multi-peak Ornstein-Uhlenbeck model). The authors found that under either model, group-living carnivores had bacula evolving toward simple shapes with intense random fluctuations around the optimum, while socially-living carnivores had bacula evolving toward complex baculum shapes with less intense random fluctuations around the optimum. Baculum length was related to body mass but not residual testis mass (although, in the results section, there was a mistake in the reporting of these results, see comments below). Baculum complexity was not related to either body mass or testis mass. The authors compared models assuming independent evolution of testis mass and length & complexity to models assuming co-evolution of testes mass with baculum length & complexity. They found neither length nor complexity coevolved with normalized testes mass. Tips and proximal ends of bacula were significantly more complex than baculum shafts but not significantly different from each other. Finally, tip complexity was weakly, positively related to intromission duration.

In general, I think this was a very interesting study that will make a valuable contribution to the literature. The authors made use of a recently developed method and overcame some challenges to doing this kind of comparative study. I think they did a good job situating their study within the literature and discussing what still needs to be done to understand baculum evolution. I have some suggestions for improving the manuscript.

The definition of complexity in biology research is often pretty slippery. Here, it's the number of simpler shapes comprising a bigger shape, and that idea can seem kind of recursive without explaining what the primitive shapes in alluded to in lines 57 and 58 are. But more generally, the word "complex" is used later in the manuscript to refer to individual features like "elaborate tips, urethral grooves", lines 66-67) whose addition to the hypothetical simple baculum makes them complex (maybe this is just a wording problem that can be fixed by changing the phrase "'complex' baculum features' to "features of complex bacula"). I think the authors should be careful to define "complexity" and then use it consistently throughout the manuscript.

Relatedly, my other big item of interest in this manuscript is about the measure of complexity used here and the inclusion of felids in the study. I think the inclusion of the felids is illustrative of the issue of defining biological complexity because, looking at the baculum shapes in Figure 2, the *Panthera leo* baculum is the odd shape out. The measure of alpha complexity returns the felid bacula as among the most complex, but I think to a human observer, the felid baculum looks more disorganized than complex, and as an outsider to the study of carnivores, I found myself wondering whether the baculum was more of a residual or less developed trait in the cats. The authors' discussion of the historical exclusion of the felids in studies of bacula (e.g., lines 356-358) suggests other people studying carnivores have had the same thought. I think a quantitative measure of complexity is an excellent thing to strive for, and probably a lot more valid than whether a shape "seems complex" to a human observer, but it does make me wonder what the difference between complexity and chaos is in describing these shapes. Here the authors used one individual of each species - how much variation is there in alpha complexity across individuals of the same felid species? How do the results of this study hold up if the felids are removed from the analysis? It seems potentially consequential to me that this group has the highest complexity measured in the study.

Additionally, the authors have used both social system and residual testes mass as proxies for the strength of postcopulatory sexual selection. The latter is likely to be closer than the former, since, as the authors point out, socially monogamous species sometimes show lots of extrapair copulations. It seems curious to me that there should be an effect of social system but not residual testes mass, if both are proxies for the same postcopulatory sexual selection, and the authors indicate they think testes mass might be a more direct measure. The authors describe in their methods that the sample size for the testes masses was 50 and that some of the measurements included epididymis mass. Given the effect of social system on baculum complexity, might the lack of relationship between residual testes mass and baculum complexity reflect the lower sample size and/or noise in the data caused by different measurement methods that the authors

discuss in lines 406-407?

Finally, I wonder if the title should be amended somewhat to reflect the mixed evidence based on different proxies for postcopulatory sexual selection and the direction of the effect of social system on baculum complexity. The title states "Postcopulatory sexual selection drives the evolution of shape complexity in the carnivoran baculum" but the authors say of their results regarding baculum complexity and testes mass in lines 398-399 that "3D shape complexity as estimated across the entire baculum is not evolving in response to post-copulatory sexual selection". It seems that the evidence from the social systems analysis shows, if anything, that postcopulatory sexual selection drives the evolution of shape simplicity.

Minor comments:

Line 30: conduct -> conducted

Line 41: This line leads me to believe the paper will report on the causative link between baculum shape and reproductive output.

Lines 56-58: This sentence is confusing and should be checked for clarity.

Lines 58-60, about 2D shape complexity of intromittent organs of water striders, doesn't seem to fit in the flow of the preceding and following sentences.

Line 61: The authors should consider stating a goal about how they're going to relate that complexity to postcopulatory sexual selection.

Line 66: There is a typo in the word "polygynandrous".

Line 73: Consider elaborating/clarifying this sentence about why expenditure in sperm competitive traits should increase with sperm competition risk because it's not crystal clear why, if it's expensive, animals should do more of it.

Line 76: Postcopulatory selection on baculum shapes may also be mediated by female stimulation (i.e., cryptic female choice/induced ovulation).

Line 80: "bacula tip" -> "baculum tip"

Line 113: "hole-filled" could mean either that the holes have been filled or that the thing is filled with holes. Consider re-phrasing to avoid ambiguity.

Line 135: How many of the body masses were sourced from the literature versus from the data that accompanied the specimens?

Line 141-142: How many species were assigned a social system based on a congener?

Line 148: Can the authors provide a reference for the mass assumption (or was this done by calculating from known masses and volumes?) because a cubic meter of testis only massing 1 kg seems surprisingly light to me.

Line 152: Please cite some papers to back up the idea that there are species increasingly recognized as both induced and spontaneous ovulators.

Line 307-309: Table 2 shows that, for the relationship between baculum length and testes mass, $p = 0.051$, so it's not appropriate here to report it as significant. I assume this is a typo - perhaps the authors meant to report on the significant relationship between body mass and baculum length.

Line 312: Consider spelling out that you looked at the relationships between testes size and baculum length as well as between testes size and baculum complexity instead of using this length/complexity notation.

Line 362: I thought the hollow areas were filled prior to analysis so I'm unclear about how this is working – perhaps the word “hollow” is misleading? Please clarify, because I think the quantification of the complexity of the felid baculum is pretty important to how the results are being interpreted and will be of great interest.

Line 458-460: Please elaborate a bit – it's not immediately clear how baculum robustness to bending (is that the same as robustness as a measure of shape and size?) clearly supports the results in the present study.

Tables and Figures:

Table 2: I see that testes mass and body mass have the same beta and SE for baculum length but they have different p-values and confidence intervals. It struck me as odd, but maybe I'm misinterpreting – please check for errors.

Figure 1: Can the authors include an inset image for the two diamonds, which show the shape at its optimal alpha value? I think the reader expects to see what the shapes faithfully reproduced look like. Otherwise we have to kind of try to interpolate between images.

Figure 2: This is a nice-looking figure! In the caption, please indicate how the uninitiated reader should identify which end of the baculum is the distal tip. Also, how did you choose which species' bacula to include here? Some indication of that might help in the caption.

Review form: Reviewer 3 (Anna Nele Herdina)

Recommendation

Accept with minor revision (please list in comments)

Scientific importance: Is the manuscript an original and important contribution to its field?

Excellent

General interest: Is the paper of sufficient general interest?

Excellent

Quality of the paper: Is the overall quality of the paper suitable?

Excellent

Is the length of the paper justified?

Yes

Should the paper be seen by a specialist statistical reviewer?

Yes

Do you have any concerns about statistical analyses in this paper? If so, please specify them explicitly in your report.

No

It is a condition of publication that authors make their supporting data, code and materials available - either as supplementary material or hosted in an external repository. Please rate, if applicable, the supporting data on the following criteria.

Is it accessible?

Yes

Is it clear?

Yes

Is it adequate?

Yes

Do you have any ethical concerns with this paper?

No

Comments to the Author

Dear Authors,

“Postcopulatory sexual selection drives the evolution of shape complexity in the carnivoran baculum” is an excellent manuscript, tackling the problem of baculum complexity. The use of alpha complexity to quantify shape complexity is innovative and will be interesting to a more general audience. This manuscript will be very useful for researchers working on genital function, reproductive biology, phylogeny, and development, as well as for evolutionary studies spanning the carnivores or mammals as a whole. It underlines that museum collections are a treasure trove of material for evolutionary research, that should be studied and valued more. This is a well and concisely written manuscript, supported by proper citations of the relevant literature (see comments for minor typos). I especially appreciate the comprehensive discussion of the findings, which also highlights how much data is still lacking on mammalian testes mass, social systems, and mating behaviour. In conclusion, I can recommend minor revisions and would like to see the following specific comments addressed:

Specific comments

If you compare baculum shape complexity without taking absolute and relative (to penis) size into account, small features on small bones will be exaggerated. The complexity of the baculum in felids (page 23, lines 356-365), is probably at least partly due to structures at the proximal end, that exist in other bacula as well. If you look at just the attachment sites to the corpora cavernosa of relatively larger bacula in the same magnification and detail, you will probably get similar complexity. The connection to the corpora cavernosa is an entheses, characterized by a calcified fibrocartilaginous layer (page 26, lines 426-429; Kelly 2000) and the attachment of collagen fibre bundles stretching from the tunica albuginea deep into the bone (Herdina et al. 2015). Entheses are sites of stress concentration and necessary for stress dissipation at the region where tendons and ligaments attach to bone (Benjamin et al. 2006). Due to the nature of such attachment sites, they need to be a microstructural mosaic of calcified features inclusive of fossae, tuberosities, crests, and ridges, as part of an adaptive osteogenic response to dissipate forces of localized mechanical loading (Waghray et al. 2015, Walters et al. 2019). This scaling effect should be addressed in the discussion.

Concerning correlations of baculum shape and size to testes mass (page 24, lines 391-399), with just one baculum per species, there could be an effect of intraspecific variability due to age, reproductive state, and possibly mating experience of the specimen. Some species show more intraspecific variability in baculum size and shape than others (Malecha et al. 2009). Depending on the specimen used, this could influence the outcome of the interspecific analyses. The width of the proximal end of the baculum seems to be more relevant than the length in intraspecific comparison (Stockley et al. 2013). I realize that an analysis with several specimens per species is out of the scope of this manuscript, but it would be an interesting future study to test this on a

subset of species.

A baculum occurs in many, but by no means most (page 3, line 38) mammalian species, which is also how it is stated in Schultz et al. 2016. Of the two main morphological types of mammalian penes, musculocavernous (e.g. in carnivores) and fibroelastic (e.g. in most artiodactyls and cetaceans), a baculum occurs only in the musculocavernous (vascular) type (Meisenheimer 1921, Schimming & Moraes 2018).

In the methods, you are referring to a different paper and its electronic supplement for CT methods and detailed scanning parameters. Please give at least a brief summary (including mean and range of resolution as voxel size) in this manuscript (page 5, line 92).

Of the 73 bacula scanned, only 50 (where testes mass was available) were used in the analysis of co-evolution between baculum length and shape complexity and normalised testes mass. Please list the species not used in this analysis or indicate them in the table of specimens in your supplement.

Minor typo editing

Page 5, line 87: or belonging to

P. 5, l. 89: micro computed tomography (CT)

P. 6, l. 111: that of the

P. 26, l. 427: paired corpora cavernosa

With kind regards,

Anna Nele Herdina

Benjamin, M., Toumi, H., Ralphs, J. R., Bydder, G., Best, T. M., & Milz, S. (2006). Where tendons and ligaments meet bone: attachment sites ('entheses') in relation to exercise and/or mechanical load. *Journal of anatomy*, 208(4), 471-490.

Herdina, A. N., Plenk Jr, H., Benda, P., Lina, P. H., Herzig-Straschil, B., Hilgers, H., & Metscher, B. D. (2015). Correlative 3D-imaging of P ipistrellus penis micromorphology: Validating quantitative microCT images with undecalcified serial ground section histomorphology. *Journal of morphology*, 276(6), 695-706.

Kelly, D. A. (2000). Anatomy of the baculum–corpus cavernosum interface in the Norway rat (*Rattus norvegicus*), and implications for force transfer during copulation. *Journal of Morphology*, 244(1), 69-77.

Malecha, A. W., Krawczyk, A. J., & Hromada, M. (2009). Morphological variability of baculum (os penis) in the polecat *Mustela putorius*. *Acta Zoologica Cracoviensia-Series A: Vertebrata*, 52(1-2), 115-120.

Meisenheimer, J. (1921). *Geschlecht und Geschlechter im Tierreiche* (Vol. 1). Fischer.

Schimming, B. C., & Moraes, G. N. (2018). Morphological analysis of the elastic and collagen fibers in the ram penis. *Pesquisa Veterinária Brasileira*, 38(11), 2159-2165.

Schultz, N. G., Lough-Stevens, M., Abreu, E., Orr, T., & Dean, M. D. (2016). The baculum was gained and lost multiple times during mammalian evolution. *Integrative and comparative biology*, 56(4), 644-656.

Stockley, P., Ramm, S. A., Sherborne, A. L., Thom, M. D., Paterson, S., & Hurst, J. L. (2013). Baculum morphology predicts reproductive success of male house mice under sexual selection. *BMC biology*, 11(1), 66.

Waghray, N., Jyothi, G. A., Imran, M., Yaseen, S., & Chaudhary, U. (2015). Enthesis: a brief review. *Apollo Medicine*, 12(1), 32-38.

Walters, M., Crew, M., & Fyfe, G. (2019). Bone Surface Micro-Topography at Craniofacial Entheses: Insights on Osteogenic Adaptation at Muscle Insertions. *The Anatomical Record*, 302(12), 2140-2155.

Decision letter (RSPB-2020-0776.R0)

27-May-2020

Dear Dr Brassey:

I am writing to inform you that your manuscript RSPB-2020-0776 entitled "Postcopulatory sexual selection drives the evolution of shape complexity in the carnivoran baculum" has, in its current form, been rejected for publication in *Proceedings B*.

This action has been taken on the advice of referees, who have recommended that substantial revisions are necessary. With this in mind we would be happy to consider a resubmission, provided the comments of the referees are fully addressed. However please note that this is not a provisional acceptance.

The three referees and Associate Editor broadly agree that the paper has merit. There is also broad agreement that the issue of variation must be dealt with somehow. But as noted, this may not be feasible during the pandemic, so there is potential flexibility-- however, some analytical way to address this issue seems still needed, not just rewording.

Please note that this decision may (or may not) have taken into account confidential comments.

In your revision process, please take a second look at how open your science is; our policy is that ***ALL*** (maximally inclusive) data involved with the study should be made openly accessible,

fully enabling re-use, replication and transparency-- see:

<https://royalsociety.org/journals/ethics-policies/data-sharing-mining/>

Insufficient sharing of data can delay or even cause rejection of a paper.

Full data and code/scripts to enable reuse/replication/repurposing are what this policy intends.

Sincerely,

Dr John Hutchinson, Editor

Associate Editor

Board Member: 1

Comments to Author:

Thank you for the opportunity to review this manuscript. Reviews from three referees have now been received. All three referees viewed the MS to address an important topic likely to be of wide interest, and generally complimented the presentation of the study. However, all three noted a number of substantive concerns, three of which were largely similar across all three reviews.

First, all three reviews expressed a major concern with regard to the characterization of baculum morphology of each species based on a single specimen. As Referee 3 notes, a full examination of variation in each of the species examined might not be feasible. However, given some cited evidence for variation in at least some species, the potential impact of such intraspecific variation on the conclusions of this study needs to be addressed. This may require more detailed assessment of variation in a subsample of taxa, requiring additional analysis or, potentially, some additional data collection.

Second, concerns were raised (in multiple contexts) about the impact of including felids in the analysis. The nature of the complexity measured in felids was questioned, with a suggestion that it may reflect disorganization reflecting a structure undergoing evolutionary reduction, rather than complexity that arose from similar drivers to that seen in other taxa. Again, some additional data analysis (e.g. jack-knifing) was suggested to address this concern.

Third, concern was expressed over whether the sample of species diversity was sufficient. In particular, some example species were identified that were not included, but which seem to show a different correspondence between baculum complexity and mating system than that identified in the study. Consideration of these concerns may also require some additional data collection and analysis.

Considering these issues, and the other specific points noted in the reviews, I cannot recommend this manuscript for publication in its current form. However, if you feel you can address the points noted in the referee reports, it may be possible to resubmit the manuscript for re-review. In a resubmission, please carefully consider the comments provided in the reviews, and explain how your revisions have addressed the concerns that were raised.

Thank you once again for your submission. I hope you find the referee comments to provide constructive guidance for revising your report on your study.

Reviewer(s)' Comments to Author:

Referee: 1

Comments to the Author(s)

Brassey et al. Postcopulatory sexual selection drives the evolution of shape complexity in the carnivoran baculum. This is an interesting paper and topic. The paper is very well written, and the approaches seem reasonable. The authors are very candid about difficulties of the study,

including how to score induced vs. spontaneous ovulators. They are also very candid in presenting results that were contrary to some of their predictions (this should be commended).

In general, however, I think this dataset needs to be expanded in order to validate the conclusions.

1) I recognize that defining complexity is difficult, but some biological meaning is missing here. Under their definition, felid bacula are considered “most complex”. However, this is driven by the fact that felid bacula are full of vacuities (as they discuss). These vacuities, in combination with the extremely small size and overall amorphous lumpiness of felid bacula suggest they are non-functional, perhaps even on an evolutionary trajectory of being lost. At the very least, felid bacula are probably not subject to the same selective pressures that other carnivoran bacula experience. Complexity can arise for a diverse number of methodological reasons, without biological function.

2) I am concerned about the small number of species analyzed. Let’s take what appears to be two extremes on the complexity scale: felids which are most complex, and pinnipeds which are least complex. All felids are scored as “S” social system, and all pinnipeds are scored “GL” (their supplementary material). A major conclusion is that S species have more complex bacula and GL species are less complex, but this could be entirely driven by just these two groups. What is most concerning is that there are plenty of pinniped species with much more complex bacula (i.e., *Zalophus*) that would be scored GL and thus reduce the strength of the correlation to social system. The larger issue is that there are many more species that should be readily available from natural history museums (raccoons, skunks, etc.). I recommend expanding the number of species to validate the inferences.

3) At the very least, the authors need to explore if/how their results change after subsampling (i.e., jackknifing) their data. For example, felids are deemed “complex” and also induced ovulators. They represent a disproportionate number of species in the dataset, so are they driving the correlation between tip complexity and induced ovulation? Or between social system and complexity? Do the results hold after removing felids? What about removing other groups? The main question is how stable are these results as different groups are removed from the dataset?

4) It is difficult to interpret patterns of interspecific divergence without replicate individuals within a species. The current dataset has sampled one baculum per species, would be good to include multiple representatives per species whenever possible.

Minor:

1) For the phylogenetic regressions between complexity, length, and testes mass – doesn’t it make more sense to use testes mass residuals?

2) Line 334 – testing differences between sections of the baculum... were these phylogenetically controlled tests?

Referee: 2

Comments to the Author(s)

The authors use a recently developed measure of shape complexity to examine baculum shapes in carnivores. They find a strong phylogenetic signal in baculum complexity across carnivores. Different social systems showed evidence of different 3D shape trait optima (θ). Two kinds of models best fit the data (Multi-regime Brownian motion model and multi-peak Ornstein-Uhlenbeck model). The authors found that under either model, group-living carnivores had bacula evolving toward simple shapes with intense random fluctuations around the optimum, while socially-living carnivores had bacula evolving toward complex baculum shapes with less intense random fluctuations around the optimum. Baculum length was related to body mass but not residual testis mass (although, in the results section, there was a mistake in the reporting of these results, see comments below). Baculum complexity was not related to either body mass or testis mass. The authors compared models assuming independent evolution of testis mass and length & complexity to models assuming co-evolution of testes mass with baculum length &

complexity. They found neither length nor complexity coevolved with normalized testes mass. Tips and proximal ends of bacula were significantly more complex than baculum shafts but not significantly different from each other. Finally, tip complexity was weakly, positively related to intromission duration.

In general, I think this was a very interesting study that will make a valuable contribution to the literature. The authors made use of a recently developed method and overcame some challenges to doing this kind of comparative study. I think they did a good job situating their study within the literature and discussing what still needs to be done to understand baculum evolution. I have some suggestions for improving the manuscript.

The definition of complexity in biology research is often pretty slippery. Here, it's the number of simpler shapes comprising a bigger shape, and that idea can seem kind of recursive without explaining what the primitive shapes alluded to in lines 57 and 58 are. But more generally, the word "complex" is used later in the manuscript to refer to individual features like "elaborate tips, urethral grooves", (lines 66-67) whose addition to the hypothetical simple baculum makes them complex (maybe this is just a wording problem that can be fixed by changing the phrase "'complex' baculum features" to "features of complex bacula"). I think the authors should be careful to define "complexity" and then use it consistently throughout the manuscript.

Relatedly, my other big item of interest in this manuscript is about the measure of complexity used here and the inclusion of felids in the study. I think the inclusion of the felids is illustrative of the issue of defining biological complexity because, looking at the baculum shapes in Figure 2, the *Panthera leo* baculum is the odd shape out. The measure of alpha complexity returns the felid bacula as among the most complex, but I think to a human observer, the felid baculum looks more disorganized than complex, and as an outsider to the study of carnivores, I found myself wondering whether the baculum was more of a residual or less developed trait in the cats. The authors' discussion of the historical exclusion of the felids in studies of bacula (e.g., lines 356-358) suggests other people studying carnivores have had the same thought. I think a quantitative measure of complexity is an excellent thing to strive for, and probably a lot more valid than whether a shape "seems complex" to a human observer, but it does make me wonder what the difference between complexity and chaos is in describing these shapes. Here the authors used one individual of each species - how much variation is there in alpha complexity across individuals of the same felid species? How do the results of this study hold up if the felids are removed from the analysis? It seems potentially consequential to me that this group has the highest complexity measured in the study.

Additionally, the authors have used both social system and residual testes mass as proxies for the strength of postcopulatory sexual selection. The latter is likely to be closer than the former, since, as the authors point out, socially monogamous species sometimes show lots of extrapair copulations. It seems curious to me that there should be an effect of social system but not residual testes mass, if both are proxies for the same postcopulatory sexual selection, and the authors indicate they think testes mass might be a more direct measure. The authors describe in their methods that the sample size for the testes masses was 50 and that some of the measurements included epididymis mass. Given the effect of social system on baculum complexity, might the lack of relationship between residual testes mass and baculum complexity reflect the lower sample size and/or noise in the data caused by different measurement methods that the authors discuss in lines 406-407?

Finally, I wonder if the title should be amended somewhat to reflect the mixed evidence based on different proxies for postcopulatory sexual selection and the direction of the effect of social system on baculum complexity. The title states "Postcopulatory sexual selection drives the evolution of shape complexity in the carnivoran baculum" but the authors say of their results regarding baculum complexity and testes mass in lines 398-399 that "3D shape complexity as estimated across the entire baculum is not evolving in response to post-copulatory sexual

selection". It seems that the evidence from the social systems analysis shows, if anything, that postcopulatory sexual selection drives the evolution of shape simplicity.

Minor comments:

Line 30: conduct -> conducted

Line 41: This line leads me to believe the paper will report on the causative link between baculum shape and reproductive output.

Lines 56-58: This sentence is confusing and should be checked for clarity.

Lines 58-60, about 2D shape complexity of intromittent organs of water striders, doesn't seem to fit in the flow of the preceding and following sentences.

Line 61: The authors should consider stating a goal about how they're going to relate that complexity to postcopulatory sexual selection.

Line 66: There is a typo in the word "polygynandrous".

Line 73: Consider elaborating/clarifying this sentence about why expenditure in sperm competitive traits should increase with sperm competition risk because it's not crystal clear why, if it's expensive, animals should do more of it.

Line 76: Postcopulatory selection on baculum shapes may also be mediated by female stimulation (i.e., cryptic female choice/induced ovulation).

Line 80: "bacula tip" -> "baculum tip"

Line 113: "hole-filled" could mean either that the holes have been filled or that the thing is filled with holes. Consider re-phrasing to avoid ambiguity.

Line 135: How many of the body masses were sourced from the literature versus from the data that accompanied the specimens?

Line 141-142: How many species were assigned a social system based on a congener?

Line 148: Can the authors provide a reference for the mass assumption (or was this done by calculating from known masses and volumes?) because a cubic meter of testis only massing 1 kg seems surprisingly light to me.

Line 152: Please cite some papers to back up the idea that there are species increasingly recognized as both induced and spontaneous ovulators.

Line 307-309: Table 2 shows that, for the relationship between baculum length and testes mass, $p = 0.051$, so it's not appropriate here to report it as significant. I assume this is a typo - perhaps the authors meant to report on the significant relationship between body mass and baculum length.

Line 312: Consider spelling out that you looked at the relationships between testes size and baculum length as well as between testes size and baculum complexity instead of using this length/complexity notation.

Line 362: I thought the hollow areas were filled prior to analysis so I'm unclear about how this is working - perhaps the word "hollow" is misleading? Please clarify, because I think the quantification of the complexity of the felid baculum is pretty important to how the results are being interpreted and will be of great interest.

Line 458-460: Please elaborate a bit – it’s not immediately clear how baculum robustness to bending (is that the same as robustness as a measure of shape and size?) clearly supports the results in the present study.

Tables and Figures:

Table 2: I see that testes mass and body mass have the same beta and SE for baculum length but they have different p-values and confidence intervals. It struck me as odd, but maybe I’m misinterpreting – please check for errors.

Figure 1: Can the authors include an inset image for the two diamonds, which show the shape at its optimal alpha value? I think the reader expects to see what the shapes faithfully reproduced look like. Otherwise we have to kind of try to interpolate between images.

Figure 2: This is a nice-looking figure! In the caption, please indicate how the uninitiated reader should identify which end of the baculum is the distal tip. Also, how did you choose which species’ bacula to include here? Some indication of that might help in the caption.

Referee: 3

Comments to the Author(s)

Dear Authors,

“Postcopulatory sexual selection drives the evolution of shape complexity in the carnivoran baculum” is an excellent manuscript, tackling the problem of baculum complexity. The use of alpha complexity to quantify shape complexity is innovative and will be interesting to a more general audience. This manuscript will be very useful for researchers working on genital function, reproductive biology, phylogeny, and development, as well as for evolutionary studies spanning the carnivores or mammals as a whole. It underlines that museum collections are a treasure trove of material for evolutionary research, that should be studied and valued more. This is a well and concisely written manuscript, supported by proper citations of the relevant literature (see comments for minor typos). I especially appreciate the comprehensive discussion of the findings, which also highlights how much data is still lacking on mammalian testes mass, social systems, and mating behaviour. In conclusion, I can recommend minor revisions and would like to see the following specific comments addressed:

Specific comments

If you compare baculum shape complexity without taking absolute and relative (to penis) size into account, small features on small bones will be exaggerated. The complexity of the baculum in felids (page 23, lines 356-365), is probably at least partly due to structures at the proximal end, that exist in other bacula as well. If you look at just the attachment sites to the corpora cavernosa of relatively larger bacula in the same magnification and detail, you will probably get similar complexity. The connection to the corpora cavernosa is an entheses, characterized by a calcified fibrocartilaginous layer (page 26, lines 426-429; Kelly 2000) and the attachment of collagen fibre bundles stretching from the tunica albuginea deep into the bone (Herdina et al. 2015). Entheses are sites of stress concentration and necessary for stress dissipation at the region where tendons and ligaments attach to bone (Benjamin et al. 2006). Due to the nature of such attachment sites, they need to be a microstructural mosaic of calcified features inclusive of fossae, tuberosities, crests, and ridges, as part of an adaptive osteogenic response to dissipate forces of localized mechanical loading (Waghray et al. 2015, Walters et al. 2019). This scaling effect should be addressed in the discussion.

Concerning correlations of baculum shape and size to testes mass (page 24, lines 391-399), with just one baculum per species, there could be an effect of intraspecific variability due to age,

reproductive state, and possibly mating experience of the specimen. Some species show more intraspecific variability in baculum size and shape than others (Malecha et al. 2009). Depending on the specimen used, this could influence the outcome of the interspecific analyses. The width of the proximal end of the baculum seems to be more relevant than the length in intraspecific comparison (Stockley et al. 2013). I realize that an analysis with several specimens per species is out of the scope of this manuscript, but it would be an interesting future study to test this on a subset of species.

A baculum occurs in many, but by no means most (page 3, line 38) mammalian species, which is also how it is stated in Schultz et al. 2016. Of the two main morphological types of mammalian penes, musculocavernous (e.g. in carnivores) and fibroelastic (e.g. in most artiodactyls and cetaceans), a baculum occurs only in the musculocavernous (vascular) type (Meisenheimer 1921, Schimming & Moraes 2018).

In the methods, you are referring to a different paper and its electronic supplement for CT methods and detailed scanning parameters. Please give at least a brief summary (including mean and range of resolution as voxel size) in this manuscript (page 5, line 92).

Of the 73 bacula scanned, only 50 (where testes mass was available) were used in the analysis of co-evolution between baculum length and shape complexity and normalised testes mass. Please list the species not used in this analysis or indicate them in the table of specimens in your supplement.

Minor typo editing

Page 5, line 87: or belonging to

P. 5, l. 89: micro computed tomography (CT)

P. 6, l. 111: that of the

P. 26, l. 427: paired corpora cavernosa

With kind regards,

Anna Nele Herdina

Benjamin, M., Toumi, H., Ralphs, J. R., Bydder, G., Best, T. M., & Milz, S. (2006). Where tendons and ligaments meet bone: attachment sites ('entheses') in relation to exercise and/or mechanical load. *Journal of anatomy*, 208(4), 471-490.

Herdina, A. N., Plenk Jr, H., Benda, P., Lina, P. H., Herzig-Straschil, B., Hilgers, H., & Metscher, B. D. (2015). Correlative 3D-imaging of P ipistrellus penis micromorphology: Validating quantitative microCT images with undecalcified serial ground section histomorphology. *Journal of morphology*, 276(6), 695-706.

Kelly, D. A. (2000). Anatomy of the baculum–corpus cavernosum interface in the Norway rat (*Rattus norvegicus*), and implications for force transfer during copulation. *Journal of Morphology*, 244(1), 69-77.

Malecha, A. W., Krawczyk, A. J., & Hromada, M. (2009). Morphological variability of baculum (os penis) in the polecat *Mustela putorius*. *Acta Zoologica Cracoviensia-Series A: Vertebrata*, 52(1-2), 115-120.

Meisenheimer, J. (1921). *Geschlecht und Geschlechter im Tierreiche* (Vol. 1). Fischer.

Schimming, B. C., & Moraes, G. N. (2018). Morphological analysis of the elastic and collagen fibers in the ram penis. *Pesquisa Veterinária Brasileira*, 38(11), 2159-2165.

Schultz, N. G., Lough-Stevens, M., Abreu, E., Orr, T., & Dean, M. D. (2016). The baculum was gained and lost multiple times during mammalian evolution. *Integrative and comparative biology*, 56(4), 644-656.

Stockley, P., Ramm, S. A., Sherborne, A. L., Thom, M. D., Paterson, S., & Hurst, J. L. (2013). Baculum morphology predicts reproductive success of male house mice under sexual selection. *BMC biology*, 11(1), 66.

Waghray, N., Jyothi, G. A., Imran, M., Yaseen, S., & Chaudhary, U. (2015). Enthesis: a brief review. *Apollo Medicine*, 12(1), 32-38.

Walters, M., Crew, M., & Fyfe, G. (2019). Bone Surface Micro-Topography at Craniofacial Entheses: Insights on Osteogenic Adaptation at Muscle Insertions. *The Anatomical Record*, 302(12), 2140-2155.

Author's Response to Decision Letter for (RSPB-2020-0776.R0)

See Appendix A.

RSPB-2020-1883.R0

Review form: Reviewer 1

Recommendation

Accept with minor revision (please list in comments)

Scientific importance: Is the manuscript an original and important contribution to its field?

Excellent

General interest: Is the paper of sufficient general interest?

Excellent

Quality of the paper: Is the overall quality of the paper suitable?

Excellent

Is the length of the paper justified?

Yes

Should the paper be seen by a specialist statistical reviewer?

No

Do you have any concerns about statistical analyses in this paper? If so, please specify them explicitly in your report.

No

It is a condition of publication that authors make their supporting data, code and materials available - either as supplementary material or hosted in an external repository. Please rate, if applicable, the supporting data on the following criteria.

Is it accessible?

Yes

Is it clear?

Yes

Is it adequate?

Yes

Do you have any ethical concerns with this paper?

No

Comments to the Author

I was Reviewer #1 initially. This is a fantastic paper, very nicely written with clear hypotheses. The results are novel (and somewhat surprising) and of broad appeal. The authors have addressed all points, although I do have some remaining.

1) In my original comment #3 I discussed jackknifing, not just felids. Specifically I asked "What about removing other groups? The main question is how stable are these results as different groups are removed from the dataset?" Yes, removing felids is important, and their removal suggests the results are robust. But what happens when you systematically remove, say, 10%, 20%, 30% of species and repeat analyses? Or systematically remove particular families? Obviously at some point all results will disappear but it would be good to understand where that threshold is.

2) In response to my original comment #2, the authors state "The current dataset stands at ~30% of extant carnivores". But of course this is not a random 30% (i.e., museum representation might not be random). Some language describing how non-random it is would be useful. For example, how much of the phylogenetic tree in terms of branch lengths was sampled (obviously after excluding species without bacula)? How many families or deeper monophyletic groups were sampled?

3) Why not use the more modern mammal tree of Upham et al. 2019. Inferring the mammal tree: Species-level sets of phylogenies for questions in ecology, evolution, and conservation. PLOS Biology 17:e3000494. Or is this the tree that was downloaded from 10K trees? If the topologies don't differ, then of course this is a moot point.

4) No matter what the answer to #3, the exact tree, with the exact branch lengths, that was used in the current study should be uploaded as supplementary data for the sake of reproducibility.

Minor things:

1) For the more general audience, it is important to clearly define polygyny, promiscuous, polygamous, and polygynandrous.

2) Line 102: resolution ranged from 7um to 60um. I assume that these different resolutions are somehow "normalized" downstream so that they don't artificially inflate size differences?

3) Just to confirm: line 226 says all data were log-transformed. This includes baculum size, body size, as well as alpha complexity?

4) Line 284: The word "outwards" took me a while to understand because I was imagining bacula coming outwards from the page. Maybe change to "outwards from tree root".

5) Table 2 is still not clear to me. Is it one model "length~testes+body" or is two models: "length~testes" and "length~body". Obviously using a single model is better.

Decision letter (RSPB-2020-1883.R0)

01-Sep-2020

Dear Dr Brassey

I am pleased to inform you that your manuscript RSPB-2020-1883 entitled "Postcopulatory sexual selection and the evolution of shape complexity in the carnivoran baculum" has been accepted for publication in *Proceedings B*. Congratulations!!

The referee(s) have recommended publication, but also suggest some moderate revisions to your manuscript. Therefore, I invite you to respond to the referee(s)' comments and revise your manuscript. Because the schedule for publication is very tight, it is a condition of publication that you submit the revised version of your manuscript within 7 days. If you do not think you will be able to meet this date please let us know. This request would surely be granted if you, as urged, do further analyses to satisfy the reviewer.

[http://datadryad.org/submit?journalID=RSPB&manu=\(Document not available\)](http://datadryad.org/submit?journalID=RSPB&manu=(Document not available)) which will take you to your unique entry in the Dryad repository. If you have already submitted your data to dryad you can make any necessary revisions to your dataset by following the above link.

Please see <https://royalsociety.org/journals/ethics-policies/data-sharing-mining/> for more details.

Sincerely,

Dr John Hutchinson

Associate Editor

Comments to Author:

Thank you for submitting your revised manuscript. The original referee that re-reviewed the MS indicated that the issues raised in review had generally been thoughtfully addressed. However, the Referee identified a few further points that would benefit from additional clarification. These involve potential consideration of additional analyses, including jackknifing of lineages other than felids, and use of an alternative phylogenetic tree of mammals. A few additional minor points were also noted.

Considering these recommendations, I encourage you to submit a revised version of the manuscript that addresses the comments provided by the referees, as well as some additional suggested corrections that I list below. Thank you once again for your submission.

L145. 'forming' should be 'form'.

L146. Change "At the stage" to "At this stage".

L149. Change "the individual" to "individual specimens".

L156. "taxa" should be "taxon"

L158. Change “and preferably those accompanied with a body mass” to “preferably accompanied by body mass”

L385. Change “highlights” to “highlight”?

L411. Change “comprising mostly of” to “being comprised mostly of”

Reviewer(s)' Comments to Author:

Referee: 1

Comments to the Author(s).

I was Reviewer #1 initially. This is a fantastic paper, very nicely written with clear hypotheses. The results are novel (and somewhat surprising) and of broad appeal. The authors have addressed all points, although I do have some remaining.

1) In my original comment #3 I discussed jackknifing, not just felids. Specifically I asked “What about removing other groups? The main question is how stable are these results as different groups are removed from the dataset?” Yes, removing felids is important, and their removal suggests the results are robust. But what happens when you systematically remove, say, 10%, 20%, 30% of species and repeat analyses? Or systematically remove particular families? Obviously at some point all results will disappear but it would be good to understand where that threshold is.

2) In response to my original comment #2, the authors state “The current dataset stands at ~30% of extant carnivores”. But of course this is not a random 30% (i.e., museum representation might not be random). Some language describing how non-random it is would be useful. For example, how much of the phylogenetic tree in terms of branch lengths was sampled (obviously after excluding species without bacula)? How many families or deeper monophyletic groups were sampled?

3) Why not use the more modern mammal tree of Upham et al. 2019. Inferring the mammal tree: Species-level sets of phylogenies for questions in ecology, evolution, and conservation. PLOS Biology 17:e3000494. Or is this the tree that was downloaded from 10K trees? If the topologies don't differ, then of course this is a moot point.

4) No matter what the answer to #3, the exact tree, with the exact branch lengths, that was used in the current study should be uploaded as supplementary data for the sake of reproducibility.

Minor things:

1) For the more general audience, it is important to clearly define polygyny, promiscuous, polygamous, and polygynandrous.

2) Line 102: resolution ranged from 7um to 60um. I assume that these different resolutions are somehow “normalized” downstream so that they don't artificially inflate size differences?

3) Just to confirm: line 226 says all data were log-transformed. This includes baculum size, body size, as well as alpha complexity?

4) Line 284: The word “outwards” took me a while to understand because I was imagining bacula coming outwards from the page. Maybe change to “outwards from tree root”.

5) Table 2 is still not clear to me. Is it one model “length~testes+body” or is two models: “length~testes” and “length~body”. Obviously using a single model is better.

Author's Response to Decision Letter for (RSPB-2020-1883.R0)

See Appendix B.

Decision letter (RSPB-2020-1883.R1)

18-Sep-2020

Dear Dr Brassey

I am pleased to inform you that your manuscript entitled "Postcopulatory sexual selection and the evolution of shape complexity in the carnivoran baculum" has been accepted for publication in Proceedings B.

Open Access

You are invited to opt for Open Access, making your freely available to all as soon as it is ready for publication under a CCBY licence. Our article processing charge for Open Access is £1700. Corresponding authors from member institutions (<http://royalsocietypublishing.org/site/librarians/allmembers.xhtml>) receive a 25% discount to these charges. For more information please visit <http://royalsocietypublishing.org/open-access>.

Paper charges

Sincerely,

Appendix A

We would like to thank the reviewers for their extremely useful feedback. We agree with the vast majority of suggestions and have revised the manuscript as such. This has been a really valuable review process, and the paper has been much improved as a result.

Thanks again,
Charlotte

Reviewer 1

Brassey et al. Postcopulatory sexual selection drives the evolution of shape complexity in the carnivoran baculum. This is an interesting paper and topic. The paper is very well written, and the approaches seem reasonable. The authors are very candid about difficulties of the study, including how to score induced vs. spontaneous ovulators. They are also very candid in presenting results that were contrary to some of their predictions (this should be commended).

We thank the reviewer for their positive feedback

In general, however, I think this dataset needs to be expanded in order to validate the conclusions.

1) I recognize that defining complexity is difficult, but some biological meaning is missing here. Under their definition, felid bacula are considered “most complex”. However, this is driven by the fact that felid bacula are full of vacuities (as they discuss). These vacuities, in combination with the extremely small size and overall amorphous lumpiness of felid bacula suggest they are non-functional, perhaps even on an evolutionary trajectory of being lost. At the very least, felid bacula are probably not subject to the same selective pressures that other carnivoran bacula experience. Complexity can arise for a diverse number of methodological reasons, without biological function.

We agree re: the strange lumpiness of felid bacula. And yes, we would also *suspect* that the felid baculum might be residual, or at least not subject to the same pressures as other carnivores. So perhaps including them ‘as if they were any other baculum’ might not be appropriate. But the tricky part is that we don’t have any real ‘functional’ data for the baculum of any carnivores, so it might also be problematic to *a priori* exclude them from an analysis based on an educated guess about their function. Or worse, by eyeballing their gross morphology, which would then make the whole analysis circular.

We’re not sure there is a clear right or wrong way to proceed. But as per the suggestions of the other reviewers as well, we’ve rerun all the analyses to exclude the cats and include that data in the Supplementary Material S3, as well as expanding on this topic in the MS. In short, it makes very little difference. The only ‘statistically significant’ change is that the correlation between baculum tip complexity and intromission duration becomes slightly insignificant (p values increases from 0.020 to 0.053). This is highlighted in the text.

The removal of felids from the analysis (as discussed above), had very little impact on the selection of best-fitting evolutionary models, or resulting parameter estimates:

Expanded dataset – with felids

Expanded dataset – without felids

Expanded dataset (w. felids) – OUMV sigma

Expanded dataset (w/o felids) – OUMV sigma

Expanded dataset (w. felids) – OUMV theta

Expanded dataset (w/o felids) – OUMV theta

2) I am concerned about the small number of species analyzed. Let's take what appears to be two extremes on the complexity scale: felids which are most complex, and pinnipeds which are least complex. All felids are scored as "S" social system, and all pinnipeds are scored "GL" (their supplementary material). A major conclusion is that S species have more complex bacula and GL species are less complex, but this could be entirely driven by just these two groups. What is most concerning is that there are plenty of pinniped species with much more complex bacula (i.e., *Zalophus*) that would be scored GL and thus reduce the strength of the correlation to social system. The larger issue is that there are many more species that should be readily available from natural history museums (raccoons, skunks, etc.). I recommend expanding the number of species to validate the inferences.

We thank the reviewer for their suggestion. The current dataset stands at ~30% of extant carnivores. Which, given the relative rarity of the baculum in museum collections, we don't believe to be too bad a sample size. However we do agree with the reviewers concern re: the potential polarity between 'complex felids' and 'simple pinnipeds' dominating the trends we find. Firstly, this has been partly addressed above by rerunning the analysis minus the felids. Secondly, we have also added 9 additional taxa to the analysis, include the *Zalophus* and raccoon suggested by the reviewer. In total, we've added a procyonid, a canid and 7 pinnipeds. As recognised by the Senior Editor, the pandemic makes access to additional collections problematic, but these samples were generated from some existing additional data we had yet to process.

We thank the reviewer for drawing our attention to the relatively more 'complex' baculum of *Zalophus*. Having added three species of eared seal to the analysis, it's notable that the group do indeed stand out as being more complex compared to the walrus and true seals. Interestingly however, having subsequently rerun all our analyses with these additional taxa, all the trends and significance levels hold, so our conclusions appear to be relatively robust. It seems there is more going on beyond just 'complex felids' vs 'simple pinnipeds'.

As evidenced above, the additional of several new taxa has had very little impact upon the selection of best-fitting evolutionary models (BMS and OUMV). Likewise, model parameter estimates are not significantly impacted by the inclusion of additional taxa, and our interpretation of the evolutionary models in the context of carnivore social systems remains the same.

Original dataset – OUMV sigma

Expanded dataset – OUMV sigma

Original dataset – OUMV theta

Expanded dataset – OUMV theta

3) At the very least, the authors need to explore if/how their results change after subsampling (i.e., jackknifing) their data. For example, felids are deemed “complex” and also induced ovulators. They represent a disproportionate number of species in the dataset, so are they driving the correlation between tip complexity and induced ovulation? Or between social system and complexity? Do the results hold after removing felids? What about removing other groups? The main question is how stable are these results as different groups are removed from the dataset?

As above, we have now rerun the analysis to remove felids and found very little impact on the overall interpretation of the patterns. And likely, we have added additional taxa (including the more complex GL pinniped taxa previously absent), and likewise this has had minimal effect.

4) It is difficult to interpret patterns of interspecific divergence without replicate individuals within a species. The current dataset has sampled one baculum per species, would be good to include multiple representatives per species whenever possible.

We thank the reviewer for their feedback. This point is likewise raised by Reviewer 3, who also recognises that intraspecific variation is interesting but beyond the scope of the current broad comparative. However we have added additional detail within the manuscript to draw further attention to this issue.

Minor:

1) For the phylogenetic regressions between complexity, length, and testes mass – doesn't it make more sense to use testes mass residuals?

As per the approach of Fitzpatrick et al (2012, Evolution) and the recommendations of Freckleton (2002), by adding body mass as a covariate in the regression: complexity ~ testes mass + body mass, we basically are testing for a correlation between complexity and testes mass residuals.

2) Line 334 – testing differences between sections of the baculum... were these phylogenetically controlled tests?

The repeated-measures ANOVA to compare tip to midshaft etc. was not phylogenetically corrected. Our regional complexity raw data strongly invalidates the assumption of sphericity in the data (i.e. assuming the variance of the differences between groups is equal) as indicated by the Mauchly Test statistic (0.77, $p < 0.0001$). This is a common issue in repeated measures ANOVA and can considerably inflate Type I error. It is therefore essential to apply a correction for sphericity, and here we use the Greenhouse-Geisser correction as implemented in the car package, as we state in-text. Whilst it would be ideal, unfortunately as far as we are aware, a mechanism for both simultaneously correcting for this sphericity and controlling for phylogenetic interrelatedness when conducted repeated-measures ANOVA does not currently exist. We have now made this clearer in-text however.

Reviewer 2

The authors use a recently developed measure of shape complexity to examine baculum shapes in carnivores. They find a strong phylogenetic signal in baculum complexity across carnivores. Different social systems showed evidence of different 3D shape trait optima (theta). Two kinds of models best fit the data (Multi-regime Brownian motion model and multi-peak Ornstein-Uhlenbeck model). The authors found that under either model, group-living carnivores had bacula evolving toward simple shapes with intense random fluctuations around the optimum, while socially-living carnivores had bacula evolving toward complex baculum shapes with less intense random fluctuations around the optimum. Baculum length was related to body mass but not residual testis mass (although, in the results section, there was a mistake in the reporting of these results, see comments below). Baculum complexity was not related to either body mass or testis mass. The authors compared models assuming independent evolution of testis mass and length & complexity to models assuming co-evolution of testes mass with baculum length & complexity. They found neither length nor complexity coevolved with normalized testes mass. Tips and proximal ends of bacula were significantly more complex than baculum shafts but not significantly different from each other. Finally, tip complexity was weakly, positively related to intromission duration.

In general, I think this was a very interesting study that will make a valuable contribution to the literature. The authors made use of a recently developed method and overcame some challenges to doing this kind of comparative study. I think they did a good job situating their study within the literature and discussing what still needs to be done to understand baculum evolution. I have some suggestions for improving the manuscript.

We thank the reviewer for their positive feedback.

The definition of complexity in biology research is often pretty slippery. Here, it's the number of simpler shapes comprising a bigger shape, and that idea can seem kind of recursive without explaining what the primitive shapes in alluded to in lines 57 and 58 are. But more generally, the word "complex" is used later in the manuscript to refer to individual features like "elaborate tips, urethral grooves", lines 66-67) whose addition to the hypothetical simple baculum makes them complex (maybe this is just a wording problem that can be fixed by changing the phrase "complex"

baculum features' to "features of complex bacula"). I think the authors should be careful to define "complexity" and then use it consistently throughout the manuscript.

Yes, we very much agree! Complexity is indeed a slippery concept. We include the example of 'the number of simple shapes comprising a bigger shape' as a more intuitive introduction to the concept for more general readers. But of course, that is not strictly how the alpha shapes algorithm actually calculates complexity. We have rephrased this section to clarify that multiple different metrics exist for quantifying complexity, and then define more clearly our use of alpha complexity herein.

Relatedly, my other big item of interest in this manuscript is about the measure of complexity used here and the inclusion of felids in the study. I think the inclusion of the felids is illustrative of the issue of defining biological complexity because, looking at the baculum shapes in Figure 2, the *Panthera leo* baculum is the odd shape out. The measure of alpha complexity returns the felid bacula as among the most complex, but I think to a human observer, the felid baculum looks more disorganized than complex, and as an outsider to the study of carnivores, I found myself wondering whether the baculum was more of a residual or less developed trait in the cats. The authors' discussion of the historical exclusion of the felids in studies of bacula (e.g., lines 356-358) suggests other people studying carnivores have had the same thought. I think a quantitative measure of complexity is an excellent thing to strive for, and probably a lot more valid than whether a shape "seems complex" to a human observer, but it does make me wonder what the difference between complexity and chaos is in describing these shapes. Here the authors used one individual of each species – how much variation is there in alpha complexity across individuals of the same felid species? How do the results of this study hold up if the felids are removed from the analysis? It seems potentially consequential to me that this group has the highest complexity measured in the study.

Yes, thank you for drawing our attention to this. As per the discussion above with Reviewer 1, we have rerun all the analyses with the cats removed and our results remain broadly the same.

Additionally, the authors have used both social system and residual testes mass as proxies for the strength of postcopulatory sexual selection. The latter is likely to be closer than the former, since, as the authors point out, socially monogamous species sometimes show lots of extrapair copulations. It seems curious to me that there should be an effect of social system but not residual testes mass, if both are proxies for the same postcopulatory sexual selection, and the authors indicate they think testes mass might be a more direct measure. The authors describe in their methods that the sample size for the testes masses was 50 and that some of the measurements included epididymis mass. Given the effect of social system on baculum complexity, might the lack of relationship between residual testes mass and baculum complexity reflect the lower sample size and/or noise in the data caused by different measurement methods that the authors discuss in lines 406-407?

This is a potential. There are lots of issues associated with testes mass. In an ideal world, we wouldn't use mass at all, but instead use the actual area of seminiferous tubules within the testes given that relative 'packing' of sperm-producing material inside the testis can vary so greatly. Unfortunately, this data is really challenging to acquire, and my students have found these tissue parameters to be extremely sensitive to the freeze-thaw processes to which most zoo-museum cadavers are subject.

Instead, testes mass is a commonly-used but imperfect proxy for postcopulatory sexual selection. Testes mass for 5 of our species also included the epididymis. Whilst there is no reason to believe this inherently biases the dataset, it may indeed add some noise to the data. We have therefore rerun the BayesTraits analyses whilst excluding these taxa, and the results are provided in the Supplementary Material S3. The patterns remain the same however: baculum *length* is weakly positively correlated to residual testes mass, whilst alpha complexity is not. And there remains no evidence for coevolution between shape complexity and these parameters. We have now included a brief mention of this in-text.

Finally, I wonder if the title should be amended somewhat to reflect the mixed evidence based on different proxies for postcopulatory sexual selection and the direction of the effect of social system on baculum complexity. The title states “Postcopulatory sexual selection drives the evolution of shape complexity in the carnivoran baculum” but the authors say of their results regarding baculum complexity and testes mass in lines 398-399 that “3D shape complexity as estimated across the entire baculum is not evolving in response to post-copulatory sexual selection”. It seems that the evidence from the social systems analysis shows, if anything, that postcopulatory sexual selection drives the evolution of shape simplicity.

The title has now been amended to reflect the more mixed pattern of results to “Postcopulatory sexual selection *and* the evolution of shape complexity in the carnivore baculum”.

Minor comments:

Line 30: conduct -> conducted

Corrected

Line 41: This line leads me to believe the paper will report on the causative link between baculum shape and reproductive output.

Amended to “Recent experiment work *has found a correlation between...*”

Lines 56-58: This sentence is confusing and should be checked for clarity.

This sentence has now been amended to:

“In the context of biological shape variation, topographically more ‘complex’ shapes are those formed by combining a greater number of simple shape primitives (cubes, cylinders, spheres, tetrahedra etc.) than less complex shapes”

Lines 58-60, about 2D shape complexity of intromittent organs of water striders, doesn’t seem to fit in the flow of the preceding and following sentences.

Amended. We included this as the only (?as far as we’re aware) other example of *complexity* explicitly being calculated for animal genitalia. As per the reviewers comments above re: defining complexity, hopefully this statement is now better integrated into the manuscript.

Line 61: The authors should consider stating a goal about how they’re going to relate that complexity to postcopulatory sexual selection.

Amended. The sentence now includes “..., and relate this metric to proxies for postcopulatory sexual selection pressures”.

Line 66: There is a typo in the word “polygynandrous”.

Corrected

Line 73: Consider elaborating/clarifying this sentence about why expenditure in sperm competitive traits should increase with sperm competition risk because it’s not crystal clear why, if it’s expensive, animals should do more of it.

Agreed. This sentence has now been modified to:

“Expenditure on spermatogenesis is expected to increase with postcopulatory sperm competition risk in order to increase a male’s fertilisation success per mating”

Line 76: Postcopulatory selection on baculum shapes may also be mediated by female stimulation (i.e., cryptic female choice/induced ovulation).

Agreed. We consider the link between baculum tip shape and induced ovulation in the next hypothesis.

Line 80: “bacula tip” -> “baculum tip”

Corrected.

Line 113: “hole-filled” could mean either that the holes have been filled or that the thing is filled with holes. Consider re-phrasing to avoid ambiguity.

Corrected to “...the CT data in which holes have been filled”.

Line 135: How many of the body masses were sourced from the literature versus from the data that accompanied the specimens?

Almost all are literature values. Actually we only use one associated body mass. Wherever possible, we’ve tried to source body mass and testes mass together, in order for the ‘residual testes mass’ to be as reliable as possible. We have now reworded to make this clearer in-text.

Line 141-142: How many species were assigned a social system based on a congener?

10 species. These are indicated as ‘(6)Lukas and Clutton-Brock’ in the ‘Social systems ref’ column of the Supplementary Material. We have also included this number in-text.

Line 148: Can the authors provide a reference for the mass assumption (or was this done by calculating from known masses and volumes?) because a cubic meter of testis only massing 1 kg seems surprisingly light to me.

Originally we assigned a neutrally buoyant 1kg/m³ as ~the average human body density. However we have now tracked down testes mass and volume data for a small number of mammals (n=7) from the Supplementary Material of Iossa et al (2008), from which an average testes density of 0.81kg/m³ is calculated. So actually, it would appear testes might be slightly more buoyant than the average body tissue. In my experience, often the testes can be surrounded by quite variable amounts of adipose tissue. This can be particularly substantial in captive zoo cadavers. It probably also varies seasonally in the wild. So these density values could still be fairly sensitive to exactly how the testes are prepared during dissection. But given that this value is more justifiable than a whole-body average, we now estimate testes mass for the 11 species from which volume is known using this density of 0.81kg/m³. But this has had a very negligible effect, changing model parameters very little and has no impact upon any levels of significance claimed.

Line 152: Please cite some papers to back up the idea that there are species increasingly recognized as both induced and spontaneous ovulators.

Agreed. We have now appended two references to the end of this statement.

Line 307-309: Table 2 shows that, for the relationship between baculum length and testes mass, $p = 0.051$, so it’s not appropriate here to report it as significant. I assume this is a typo – perhaps the authors meant to report on the significant relationship between body mass and baculum length.

Yes apologies. This has now been corrected, and amended in light of the new taxa added to the analysis:

“Baculum length was found to be significantly, albeit very weakly, positively correlated with residual testes mass ($p=0.04$, Table 2), whilst baculum complexity was not ($p>0.05$)”

Line 312: Consider spelling out that you looked at the relationships between testes size and baculum length as well as between testes size and baculum complexity instead of using this length/complexity notation.

Agreed. This has been amended to:

“In addition, we tested for *co-evolution* between the carnivoran baculum and testes size by comparing the posterior distribution of models in which a) baculum *length* and normalised testes mass evolve independently of one another and b) baculum *complexity* and normalised testes mass evolve independently of one another, against models in which the two traits co-evolve”

Line 362: I thought the hollow areas were filled prior to analysis so I’m unclear about how this is working – perhaps the word “hollow” is misleading? Please clarify, because I think the quantification of the complexity of the felid baculum is pretty important to how the results are being interpreted and will be of great interest.

Yes of course. I suppose that’s a misuse of the word ‘hollow’. Any vacuities that were entirely enclosed by the periosteal contour were indeed filled prior to analysis. In the case of the proximal cat baculum, there are often paired ‘deep depressions’, but these do still form part of the bone’s external contour. We have amended to be more specific with our language.

Line 458-460: Please elaborate a bit – it’s not immediately clear how baculum robustness to bending (is that the same as robustness as a measure of shape and size?) clearly supports the results in the present study.

Agreed. In the case of our previous FEA analysis, I think the fact that we found a negative correlation between intromission duration and stress values when the bones were loaded in the ventral direction was because the species with the longest copulation times also have a dorsally-directed distal hook. In addition to increasing complexity in the present analysis, the distal hook would act to mitigate stress under dorsoventral bending when load is applied at the tip. We have now clarified this in-text.

Tables and Figures:

Table 2: I see that testes mass and body mass have the same beta and SE for baculum length but they have different p-values and confidence intervals. It struck me as odd, but maybe I’m misinterpreting – please check for errors.

We appreciate the reviewer really checking the details! Yes, thankfully that wasn’t an error. The slopes were slightly different, just both rounded to 0.16. As can be seen in the new Figure 2 with added species, the slopes are still very close but slightly different (0.15 vs 0.16).

Figure 1: Can the authors include an inset image for the two diamonds, which show the shape at its optimal alpha value? I think the reader expects to see what the shapes faithfully reproduced look like. Otherwise we have to kind of try to interpolate between images.

Good suggestion. The alpha shape representing the ‘optimal’ alpha value has now been added to the figure.

Figure 2: This is a nice-looking figure! In the caption, please indicate how the uninitiated reader should identify which end of the baculum is the distal tip. Also, how did you choose which species’ bacula to include here? Some indication of that might help in the caption.

Thank you for your feedback. Yes, we have now included this information in the figure caption.

Reviewer 3

“Postcopulatory sexual selection drives the evolution of shape complexity in the carnivoran baculum” is an excellent manuscript, tackling the problem of baculum complexity. The use of alpha complexity to quantify shape complexity is innovative and will be interesting to a more general audience. This manuscript will be very useful for researchers working on genital function, reproductive biology, phylogeny, and development, as well as for evolutionary studies spanning the carnivores or mammals as a whole. It underlines that museum collections are a treasure trove of material for evolutionary research, that should be studied and valued more. This is a well and concisely written manuscript, supported by proper citations of the relevant literature (see comments for minor typos). I especially appreciate the comprehensive discussion of the findings, which also highlights how much data is still lacking on mammalian testes mass, social systems, and mating behaviour. In conclusion, I can recommend minor revisions and would like to see the following specific comments addressed:

We thank the reviewer for their positive feedback.

Specific comments

If you compare baculum shape complexity without taking absolute and relative (to penis) size into account, small features on small bones will be exaggerated. The complexity of the baculum in felids (page 23, lines 356-365), is probably at least partly due to structures at the proximal end, that exist in other bacula as well. If you look at just the attachment sites to the corpora cavernosa of relatively larger bacula in the same magnification and detail, you will probably get similar complexity. The connection to the corpora cavernosa is an enthesis, characterized by a calcified fibrocartilaginous layer (page 26, lines 426-429; Kelly 2000) and the attachment of collagen fibre bundles stretching from the tunica albuginea deep into the bone (Herdina et al. 2015). Enteses are sites of stress concentration and necessary for stress dissipation at the region where tendons and ligaments attach to bone (Benjamin et al. 2006). Due to the nature of such attachment sites, they need to be a microstructural mosaic of calcified features inclusive of fossae, tuberosities, crests, and ridges, as part of an adaptive osteogenic response to dissipate forces of localized mechanical loading (Waghray et al. 2015, Walters et al. 2019). This scaling effect should be addressed in the discussion.

We thank the reviewer for their thought-provoking insight. For morphological features that are likely to scale with size, the methodology would of course consider such features to be equally ‘complex’. However as the reviewer also highlights, the attachment of collagen bundles etc. creates a very fine-scale fibrous surface texture at the proximal end. Interesting, the allometry of enteses (albeit from humeral tendons) suggests that microscale features of the interfacial roughness at such attachment site are indeed *size invariant* across several species (Deymier-Black et al., 2015). The degree to which these features are considered complex would therefore be expected to vary with respect to the absolute size of the baculum, assuming the baculum enthesis behaves the same. This is an interesting issue that also arises in other morphometric approaches that normalise for size. We have added a section to the discussion to explore this further.

Concerning correlations of baculum shape and size to testes mass (page 24, lines 391-399), with just one baculum per species, there could be an effect of intraspecific variability due to age, reproductive state, and possibly mating experience of the specimen. Some species show more intraspecific variability in baculum size and shape than others (Malecha et al. 2009). Depending on the specimen used, this could influence the outcome of the interspecific analyses. The width of the proximal end of the baculum seems to be more relevant than the length in intraspecific comparison (Stockley et al. 2013). I realize that an analysis with several specimens per species is out of the scope of this manuscript, but it would be an interesting future study to test this on a subset of species.

Yes absolutely. This is always the trade-off between maximising your interspecific coverage vs. adequately describing any intraspecific variation. I wouldn't expect our sample to be *systematically* biased by these issues (for example, hypothetically if we had only sampled subadult pinnipeds vs.

the rest of the sample being adult; OR if all the mustelids were sampled during the mating season vs. the rest being sampled out of season). But I agree, these factors likely to add *noise* to the morphological data. We've now added some further thoughts to this end in the Discussion.

A baculum occurs in many, but by no means most (page 3, line 38) mammalian species, which is also how it is stated in Schultz et al. 2016. Of the two main morphological types of mammalian penes, musculocavernous (e.g. in carnivores) and fibroelastic (e.g. in most artiodactyls and cetaceans), a baculum occurs only in the musculocavernous (vascular) type (Meisenheimer 1921, Schimming & Moraes 2018).

Ah yes, agreed. I suppose I was inferring from Schultz that their 925 out of 1028 subsample of species with a baculum would hold if scaled up to all ~3700 mammal species. Given that they find *all* rodents included have a baculum, plus the vast majority of bats studied, it is *likely* the case that the majority of modern mammals have a baculum. The clades with musculocavernous penes also tend to be the most speciose. But the reviewer is correct, we don't have true presence/absence data for most taxa, so we'll rephrase.

In the methods, you are referring to a different paper and its electronic supplement for CT methods and detailed scanning parameters. Please give at least a brief summary (including mean and range of resolution as voxel size) in this manuscript (page 5, line 92).

Agreed. This detail has now been added to the main text of the manuscript.

Of the 73 bacula scanned, only 50 (where testes mass was available) were used in the analysis of co-evolution between baculum length and shape complexity and normalised testes mass. Please list the species not used in this analysis or indicate them in the table of specimens in your supplement.

The species missing testes mass data are represented by a 'NA' value in the testes mass column of Supplementary Material S1. However, we now explicitly state this in-text to make it clearer.

Minor typo editing

Page 5, line 87: or belonging to

Corrected

P. 5, l. 89: micro computed tomography (CT)

Corrected

P. 6, l. 111: that of the

Corrected

P. 26, l. 427: paired corpora cavernosa

Corrected

Additional changes to the OUWIE analysis

Having returned to the 'OUWIE' R package to rerun the above analyses at the request of the reviewers, we have noted that the authors of OUWIE have made several modifications to their script in the latest version of the package. Specifically, in the first iteration of our manuscript, we set 'root.station=T' in OUWIE for the fitting of evolutionary models. However, as per the latest version of the OUWIE package (v2.3), the authors of the package have now recognized that they were incorrectly implementing this stationarity function (see Beaulieu et al., R OUWIE help guide). In the latest version, they have now correctly implemented *true* root stationarity, but only for the OU1 and OUM models. In revising the MS and rerunning the code using the latest version of OUWIE, we find this new implementation of root.station=T to be infeasibly slow to run across our 10000 trees. We have therefore run the model-fitting over 10 trees using both root.station=T and root.station=F in the new version of the package, and compared the AICc model selection protocol below:

root.station=F

root.station=T

As can be seen, the choice of best-fitting models on the basis of low AICc scores is unlikely to be affected by the root.station parameter, and we therefore proceed by dropping the root optima (root.station=F) in the present version of the manuscript.

In addition, we also removed the OUMA evolutionary model from the above analysis. In the previous analysis, we had removed any iteration (of the 10,000) in which *any* model (BM, BMS, OU1, OUMA etc.) had produced negative eigenvalues of the Hessian. However, when running the model fitting on the latest dataset, the OUMA model produced negative eigenvalues for ~95% of the iterations, suggesting parameter estimates were unreliable and the ML solution had not been found. Rather than throw away the vast majority of simulations for the other models, we choose to exclude OUMA from the evolutionary model-fitting process.

As evidenced above, the additional of several new taxa and the removal of the OUMA model has had very little impact upon the selection of best-fitting evolutionary models (BMS and OUMV). Likewise, model parameter estimates are not significantly impacted by the inclusion of additional taxa, and our interpretation of the evolutionary models in the context of carnivore social systems remains the same.

Original dataset – OUMV sigma

Expanded dataset – OUMV sigma

Original dataset – OUMV theta

Expanded dataset – OUMV theta

In addition, the previous version of OUMV erroneously produced optimal theta under the BMS model scenario that may differ between the various social systems (previously shown in our Figure 4C). In fact, only sigma values should vary between social systems in this model. This parameter is no longer calculated for BMS and has been removed from Figure 4. Our assertion that optimal baculum complexity varies with social system is still held up by the results of the well-fitted OUMV model.

Appendix B

Reply to Reviewers Comments – Brassey et al.

We thank the Reviewers and Editors for a very positive review process. We have addressed Reviewer 1's remaining concerns below.

Many thanks
Charlotte

#####

Associate Editor Comments:

Thank you for submitting your revised manuscript. The original referee that re-reviewed the MS indicated that the issues raised in review had generally been thoughtfully addressed. However, the Referee identified a few further points that would benefit from additional clarification. These involve potential consideration of additional analyses, including jackknifing of lineages other than felids, and use of an alternative phylogenetic tree of mammals. A few additional minor points were also noted.

See below for details of the jack-knifing and alternative phylogeny

Considering these recommendations, I encourage you to submit a revised version of the manuscript that addresses the comments provided by the referees, as well as some additional suggested corrections that I list below. Thank you once again for your submission.

L145. 'forming' should be 'form'.

L146. Change "At the stage" to "At this stage".

L149. Change "the individual" to "individual specimens".

L156. "taxa" should be "taxon"

L158. Change "and preferably those accompanied with a body mass" to "preferably accompanied by body mass"

L385. Change "highlights" to "highlight"?

L411. Change "comprising mostly of" to "being comprised mostly of"

All the above grammatical issues have been corrected.

#####

Reviewer 1 Comments:

I was Reviewer #1 initially. This is a fantastic paper, very nicely written with clear hypotheses. The results are novel (and somewhat surprising) and of broad appeal. The authors have addressed all points, although I do have some remaining.

We thank the reviewer for their additional thoughts. We have addressed all of their concerns below, and made the appropriate additions in-text whenever necessary.

1) In my original comment #3 I discussed jackknifing, not just felids. Specifically I asked "What about removing other groups? The main question is how stable are these results as different groups are removed from the dataset?" Yes, removing felids is important, and their removal suggests the results are robust. But what happens when you systematically remove, say, 10%, 20%, 30% of

species and repeat analyses? Or systematically remove particular families? Obviously at some point all results will disappear but it would be good to understand where that threshold is.

We thank the reviewer for their suggestion. We have taken their advice and run a random downsampling exercise, removing species from our samples in evenly-spaced increments and rerunning the analysis. As below (re: the alternative phylogeny) because some of these tests are very computationally time-consuming, the downsampling exercise has in some instances been run across a distribution of either 10 or 100 phylogenies, as opposed to the 10,000 in the main manuscript. Therefore some tests below do not cover the full degree of phylogenetic uncertainty incorporated in the main manuscript, but they do paint a reasonable picture of how sensitive/insensitive some of the tests are to downsampling.

The results are fairly unsurprising. Those statistical tests that were originally *highly* significant or *highly* insignificant remain relatively robust to a fair amount of downsampling. Those results that hover around the $p=0.05$ mark are of course more sensitive to reducing the sample size. I provide a summary of some of the downsampling tests below.

Lambda (phylogenetic signal in baculum complexity) – calculated on a consensus phylogeny. Subsampling process repeated 100 times per increment, and a mean taken. Downsampled by 5-95% in 5% increments.

OUIE evolutionary model selection – calculated across 10 phylogenies. Downsampled by 10-50% in 10% increments.

10% downsample 20% downsample 30% downsample 40% downsample 50% downsample

Beyond 20% downsampling, the model-fitting process warns there may not be enough data for reliable fitting of the OUM model. This is also the case for the OU1 model at 50% downsampling. The

preferred choice of model (those with the lowest AICc scores) remains remarkably consistent despite considerable downsampling.

OUMV evolutionary model parameters - calculated across 10 phylogenies for the OUMV model. Downsampled by 10-50% in 10% increments.

Sigma

Theta

Violin plots are used here as means of comparison to the original manuscript, but are not particularly informative as each 'violin' now only represents the 10 phylogenetic iterations over which the subsampling was conducted (hence the 'unusual' shapes!). Most importantly, the relative trends between social systems remain robust to quite heavy downsampling.

BayesTraits regression (baculum complexity ~ testes mass+body mass) – calculated across 10,000 phylogenies. Downsampled by 5-50% in 2% increments. Each increment repeated twice and averaged.

BayesTraits regression (baculum length ~ testes mass+body mass) – calculated across 10,000 phylogenies. Downsampled by 5-50% in 2% increments. Each increment repeated twice and averaged.

BayesTraits regression (baculum tip complexity ~ intromission duration or ovulation style) – calculated across 10,000 phylogenies. Downsampled by 5-50% in 2% increments. Each increment repeated twice and averaged.

Unsurprisingly, those p -values hovering around a significant level of 0.05 are relatively sensitive to downsampling. There is a high degree of scatter, in part due to each downsampling increment being repeated only twice, due to the lengthy computational time required to run said analysis. Furthermore, the downsampling operation is run on the original dataset of 82 taxa. *Subsequent* to downsampling, only those taxa for which testes mass, intromission time etc is available are then included in the regression. The percentage decrease in the sample size of the BayesTraits analysis will therefore not exactly equal the percentage downsample of the original dataset.

Undeniably, our sample size is at the lower end of that with which significant relationships can be recovered in some of the regression analyses above. However, for those trends we do consider to be 'significant' in-text, a reassuring trend is visible in which p -values do decrease with increasing sample size, suggesting these significant results are not anomalous false positives. In the main text of the manuscript, we already draw attention to the fact that these are weak, marginally significant results. In the future, the inclusion of additional taxa will likely firm up these relationships further. For those results in which we recover *no* significant relationship (baculum complexity ~ relative testes mass, for example), the trend above suggests that this result is likely to hold true even if the sample size was substantially increased.

We agreed with the Reviewer's initial suggestion to remove the felids from the original analysis and repeat. This was justified both on the grounds of their highly unusual morphology relative to the rest of the families, and the fact that other previous authors had done likewise. And we were pleased to see the removal of the felids had very little impact upon our interpretations. However, we would argue the further removal of additional clades cannot be justified on the above grounds. Whilst authors have argued that the felid baculum may be residual, no such debate has occurred for other groups of carnivore. When faced with prospect of throwing away data, we would argue there needs to be a strong logic underlying this choice. Beyond the random downsampling outlined above (which is useful for determining the effects of sample size), the decision to exclude say, canids, from the analysis based purely upon their taxonomic grouping seems arbitrary when there is no obvious functional or morphological reason to exclude them.

2) In response to my original comment #2, the authors state "The current dataset stands at ~30% of extant carnivores". But of course this is not a random 30% (i.e., museum representation might not be random). Some language describing how non-random it is would be useful. For example, how much of the phylogenetic tree in terms of branch lengths was sampled (obviously after excluding species without bacula)? How many families or deeper monophyletic groups were sampled?

In terms of % branch length covered relative to the full Carnivore tree (minus hyaenas and binturong lacking a baculum), we have 50.5% tree coverage of extant Carnivores.

In terms of the number of families, I believe there are 15 extant families of carnivores known to possess a baculum (Hyaenidae do not, as far as we're aware). We have data for 12. Those missing include the skunks, the monospecific Nandiniidae (African palm civet) and the Linsang (2 species within Prionodontitidae). We are therefore confident that we have sampled relatively well across the Carnivore phylogeny, and we went out of our way to acquire felid microCT scans of whole penes, in order to remedy their otherwise absence from museum collections.

We have added a brief sentence in-text to this effect.

3) Why not use the more modern mammal tree of Upham et al. 2019. Inferring the mammal tree: Species-level sets of phylogenies for questions in ecology, evolution, and conservation. PLOS Biology 17:e3000494. Or is this the tree that was downloaded from 10K trees? If the topologies don't differ, then of course this is a moot point.

In the spirit of honesty, we used the 10k mammal tree rather than the Upham tree published in November 2019 because we started running a lot of these analyses *before* November 2019! But I recognise that isn't a legitimate reason to not use the more modern phylogeny. Thankfully the topology of carnivores has remained remarkably consistent, and the fact that we run most of our analyses on the posterior distribution of 10,000 trees means that a degree of uncertainty in the tree is already incorporated into the analysis. Of course, the posterior distributions of the two trees will themselves likely differ. If we consider the consensus trees as being representative, then there is some shifting in the canids (*Speothus* in particular), and in the position of *Gulo gulo* in the mustelids between the two trees. I've therefore run a quick analysis to check how sensitive our results are to the choice of tree.

Using the Upham tree, calculated phylogenetic signal in optimal alpha shape complexity changed very little; ($\lambda=0.89$, $\log L=-13.08$, $p<0.001$) versus ($\lambda=0.85$, $\log L=-15.47$, $p<0.001$). Given that the topology has changed very little, we are reluctant to rerun our full suite of statistics across 10,000 trees again, as this takes ~5-7 days of computer processing time. But as an example, we have rerun the OUWIE analysis on 100 trees of the Upham phylogeny and the difference is very negligible

10k trees

VertLife Upham trees

The violin plots obviously look 'coarser' due to only using 100 vs 10,000 simulations. And whilst the absolute values for AICc and the OUWle parameters change given the new node ages and branch lengths etc., the *trends* remain remarkably consistent and do not change our interpretations at all. Given the considerable extra processing time that would be required to rerun the full analyses (with

and without felids) for 10,000 simulations, we feel it is appropriate to stick with our original phylogeny. However, we appreciate the reviewer drawing this new dataset to our attention, and we will doubtless make use of these updated phylogenies moving forward.

4) No matter what the answer to #3, the exact tree, with the exact branch lengths, that was used in the current study should be uploaded as supplementary data for the sake of reproducibility.

Yes, we agree. The consensus tree and 10k distribution have been added to the supplementary material S2 and S3.

Minor things:

1) For the more general audience, it is important to clearly define polygyny, promiscuous, polygamous, and polygynandrous.

Yes agree. We're quite 'up against the wall' with the maximum page limit of PRSB, but I've now added in a brief description in-text and direct readers to the original source for further details.

2) Line 102: resolution ranged from 7um to 60um. I assume that these different resolutions are somehow "normalized" downstream so that they don't artificially inflate size differences?

During processing of the CT data within the alpha shapes analysis, all specimens are represented by the same number of vertices (100,000) and size differences are removed by scaling the point clouds on the basis of distance between nearest neighbours. This is detailed further in the methods paper of Gardiner et al.

3) Just to confirm: line 226 says all data were log-transformed. This includes baculum size, body size, as well as alpha complexity?

Yes correct, all log₁₀ transformed.

4) Line 284: The word "outwards" took me a while to understand because I was imagining bacula coming outwards from the page. Maybe change to "outwards from tree root".

Thanks, good spot. This has been corrected.

5) Table 2 is still not clear to me. Is it one model "length~testes+body" or is two models: "length~testes" and "length~body". Obviously using a single model is better.

Yes, it's the single model, "length~testes+body". We were replicating the Table 1 layout of Fitzpatrick's 2012 pinniped baculum paper. We've added the formula into the Table legend to make this clearer.